# CHUK/IKK-α loss in lung epithelial cells enhances NSCLC growth associated with HIF up-regulation

Evangelia Chavdoula[1,2,3], David M Habiel[4] , Eugenia Roupakia[2,3], Georgios S Markopoulos[2,3] , Eleni Vasilaki[1], Antonis Kokkalis[1] , Alexander P Polyzos[1], Haralabia Boleti[5], Dimitris Thanos[1], Apostolos Klinakis[1] , Evangelos Kolettas[2,3,*] , Kenneth B Marcu[1,2,3,6,7,*]

Through the progressive accumulation of genetic and epigenetic alterations in cellular physiology, non–small-cell lung cancer (NSCLC) evolves in distinct steps involving mutually exclusive oncogenic mutations in K-Ras or EGFR along with inactivating mutations in the p53 tumor suppressor. Herein, we show two independent in vivo lung cancer models in which CHUK/IKK-α acts as a major NSCLC tumor suppressor. In a novel transgenic mouse strain, wherein IKKα ablation is induced by tamoxifen (Tmx) solely in alveolar type II (AT-II) lung epithelial cells, IKKα loss increases the number and size of lung adenomas in response to the chemical carcinogen urethane, whereas IKK-β instead acts as a tumor promoter in this same context. IKKα knockdown in three independent human NSCLC lines (independent of K-Ras or p53 status) enhances their growth as tumor xenografts in immune-compromised mice. Bioinformatics analysis of whole transcriptome profiling followed by quantitative protein and targeted gene expression validation experiments reveals that IKKα loss can result in the up-regulation of activated HIF-1-α protein to enhance NSCLC tumor growth under hypoxic conditions in vivo.

## Introduction

Lung cancer (LC) is the most common cancer and the leading cause of cancer-related deaths worldwide in males and females. Lung cancer is clinically divided into non–small-cell lung cancer (NSCLC), including adenocarcinoma, squamous cell carcinoma (SCC), and large cell carcinoma, representing ~85% and small cell lung cancer representing ~15%, of all LCs diagnosed. The prognosis of LC patients is still disappointing, with a 5-yr overall survival generally less than ~18%. Non–small-cell lung cancer, with adenocarcinoma being the major histopathologic subtype, is often intrinsically resistant to chemo- and radiotherapy, and its development involves a number of genetic and epigenetic events (Sun et al, 2007; Herbst et al, 2008; Siegel et al, 2016).

In NSCLC patients, mutually exclusive oncogenic K-Ras mutations and epidermal growth factor receptor mutations or amplifications occur in ~30% and 10–40%, respectively, whereas inactivating, mostly missense, mutations in the p53 tumor suppressor are found in >50% of cases (Ding et al, 2008; Greulich, 2010). Most K-Ras point mutations are G-T transversions in codon 12, or mutations in codons 13 and 61, which are indicative of poor prognosis for early- and late-stage NSCLC (Ding et al, 2008; Greulich, 2010). Non–small-cell lung cancer with oncogenic K-Ras mutations is refractory to pharmacological treatment targeted to Ras enzymatic activity because mutant K-Ras oncoproteins lack the normal protein's intrinsic GTPase function. However, mutated Ras–driven signaling pathways have a variety of downstream targets and are also linked to other cellular pathways amenable to drug treatment, some of which have also been found mutated or aberrantly expressed in lung tumors. Thus, it could be argued that blocking one of these downstream targets or pathways should have significant therapeutic effect (Diaz et al, 2012; Misale et al, 2012).

Transgenic mouse models have established a causal relationship between K-Ras and p53 mutations in LC (Guerra et al, 2003; Tuveson et al, 2004; Meylan et al, 2009; de Seranno & Meuwissen, 2010; Farago et al, 2012), where cancer induction by urethane (Kelly-Spratt et al, 2009) or lung-specific expression of mutant p53[273His] either accompanied by K-Ras mutations or via conditional expression of oncogenic K-Ras showed that K-Ras mutations are an initiating event in NSCLC

[1]Biomedical Research Foundation Academy of Athens, Athens, Greece    [2]Laboratory of Biology, School of Medicine, Faculty of Health Sciences, University of Ioannina, University Campus, Ioannina, Greece    [3]Biomedical Research Division, Institute of Molecular Biology and Biotechnology, Foundation for Research and Technology, Ioannina, Greece    [4]Cedars-Sinai Medical Center, Los Angeles, CA, USA    [5]Intracellular Parasitism Laboratory, Department of Microbiology and Light Microscopy Unit, Hellenic Pasteur Institute, Athens, Greece    [6]Departments of Biochemistry and Cell Biology and Pathology, Stony Brook University, Stony Brook, NY, USA    [7]Department of Biological Sciences, San Diego State University, San Diego, CA, USA

Correspondence: ekoletas@uoi.gr; kenneth.marcu@stonybrook.edu
Evangelia Chavdoula's present address is Department of Cancer Biology and Genetics, College of Medicine and Arthur G. James Comprehensive Cancer Center, The Ohio State University, Columbus, OH, USA
David M Habiel's present address is Sanofi US, Cambridge, MA, USA
Antonis Kokkalis's present address is the Broad Institute of MIT and Harvard, Boston, MA, USA and Dana Farber Cancer Institute, Boston, MA, USA
*Kenneth B Marcu and Evangelos Kolettas contributed equally to this work

development (de Seranno & Meuwissen, 2010; Farago et al, 2012). Moreover, *EGFR* and *K-Ras* mutations are mutually exclusive in NSCLC with the emergence of *K-Ras* mutations associated with resistance to EGFR-targeted cancer therapies (Diaz et al, 2012; Misale et al, 2012). Importantly, in this context, mutant *K-Ras* programming leads to inflammation (Ji et al, 2006; Moghaddam et al, 2009; Xia et al, 2012) and enhanced canonical NF-κB activity (Meylan et al, 2009; Basseres et al, 2010; Xia et al, 2012) in mouse NSCLC models. In a conditional CC10-Cre/LSL-*K-Ras*^G12D mouse LC model wherein *K-Ras* expression was targeted to Clara cells, mice developed pronounced pulmonary inflammation and lung tumors (Ji et al, 2006). A recent study showed that *K-Ras* expression induced lung adenocarcinoma and the mice displayed increased cytokine production and inflammatory cell infiltration in the bronchoalveolar lavage after tumor initiation (Xia et al, 2012).

The NF-κB transcription factors (TFs) can either activate or repress target gene transcription in different physiological contexts (Perkins, 2007, 2012; Penzo et al, 2009; Hayden, 2012). The NF-κB TFs are critical regulators of pro-inflammatory/stress-like responses; and their immediate upstream signaling components are aberrantly expressed and/or activated in pulmonary diseases, including NSCLC, and have been implicated in the unfavorable prognosis for patient survival (Greenman et al, 2007; Giopanou et al, 2015). The NF-κB TFs bind to DNA as heterodimers or homodimers of five possible subunits (RelA/p65, c-Rel, RelB, p50, and p52). All NF-κB family members contain an N-terminal Rel homology domain that mediates DNA binding and dimerization. The c-Rel, p65/RelA, and RelB subunits contain a C-terminal transactivation domain, unlike the p50 and p52 subunits (which are derived by processing of their larger precursors p105/NF-κB1 and p100/NF-κB2, respectively), which lack a transcriptional activation domain. Archetypical p65/p50 heterodimers are restrained in the cytoplasm by IκBs (NF-κB inhibitors) in most cells in the absence of stress responses. Canonical NF-κB activation requires the phosphorylation of serine residues (Ser) 32 and 36 in IκBα's signal response domain (SRD), causing IκBα ubiquitination/proteasomal degradation, resulting in NF-κB p65/50 dimer nuclear translocation and target gene activation. The IκBα SRD phosphorylation is mediated by the IKK signalosome complex (IKKα and IKKβ Ser/Thr kinases, and NEMO/IKKγ, a regulatory/adaptor protein). NEMO is required for IKKβ activation by phosphorylation of its T-activating loop Ser177/181, which occurs in response to pro-inflammatory/stress-related extracellular signals. In contrast to IKKβ, IKKα-activating phosphorylation of Ser176/180 is NEMO-independent and mediated by NF-κB–inducing kinase (NIK), which is activated by stimuli of adaptive immune responses. Furthermore, the noncanonical NF-κB mediator, p100/NF-κB2 precursor protein, functions akin to IκBα by sequestering RelB in the cytoplasm and also a subset of p65/p50 heterodimers (Karin & Greten, 2005; Perkins, 2007, 2012; Chariot, 2009; Karin, 2009; Hayden, 2012). IKKα (but not IKKβ) phosphorylates an IκB-like signal SRD in p100/NF-κB2. This phosphorylation induces p100/NF-κB2 ubiquitination/proteasome-dependent processing, yielding the mature NF-κB p52 subunit and p52/RelB heterodimers, which in turn translocate to the nucleus and activate a distinct set of noncanonical NF-κB target genes.

Canonical NF-κB signaling has been implicated in NSCLC genesis in the context of *K-Ras* activation. *K-Ras* was shown to activate canonical NF-κB in mouse LC models and human lung epithelial cells (Meylan et al, 2009; Basseres et al, 2010; Xia et al, 2012) via the generation of inflammatory responses in lung tumors (Ji et al, 2006; Iwanaga et al, 2008). Studies in vivo have used the classical urethane (Stathopoulos et al, 2007; Kelly-Spratt et al, 2009) or the conditional *K-Ras*–induced murine models of lung carcinogenesis, in which canonical NF-κB was suppressed using an IκBαSR super-repressor (Stathopoulos et al, 2007; Meylan et al, 2009) or by ablation of RelA/p65 (Basseres et al, 2010) or IKKβ (Xia et al, 2012), which led to reduced proliferation of *K-Ras*^G12D–induced lung adenocarcinomas in mice. In addition, mouse strains susceptible to lung tumor formation (FVB, BALB/c) exhibit NF-κB activation and inflammation in lungs in response to urethane for 4 wk, detected in airway and type II alveolar cells and macrophages (Stathopoulos et al, 2007). In an inducible transgenic FVB mouse model expressing an IκBαDN (dominant negative) transgene in airway epithelium, urethane-induced lung inflammation was blocked and tumor formation was reduced indicating that canonical NF-κB signaling in airway epithelium is integral to lung tumorigenesis induced by urethane (Stathopoulos et al, 2007) and may be a potential chemopreventive NSCLC target.

In contrast to the involvement of IKKβ-mediated canonical NF-κB signaling in lung inflammation, cancer development, and progression, the functional roles of IKKα in these processes remain unclear. Pancreas-specific IKKα ablation triggers spontaneous pancreatitis in mice, resembling chronic pancreatitis in humans. Indeed, IKKα was observed to be a negative regulator of inflammation (Lawrence et al, 2005; Li et al, 2005) and to maintain pancreatic homeostasis independently of NF-κB, unlike IKKβ (Li et al, 2013). Mammary tumor development in response to a carcinogen or an *MMTVc-neu* (*ErbB2*/*Her2*) transgene was retarded in *Ikkα*^S176A/S180A knock-in mouse (Cao et al, 2007; Zhang et al, 2013). Similarly, in a mouse prostate cancer model, it was shown that nuclear IKKα suppresses the metastasis suppressor maspin and promotes prostate cancer progression (Luo et al, 2007). IKKα also promotes colon cancer (Fernandez-Majada et al, 2007; Margalef et al, 2012; Goktuna et al, 2014). In contrast, the epigenetic loss of IKKα in oral carcinomas is associated with tumor progression (Maeda et al, 2007). Moreover, overexpression of IKKα induces differentiation and reduces tumorigenicity of nasopharyngeal cancer (NPC) cells without activating NF-κB signaling (Yan et al, 2014; Deng et al, 2015; Xiao et al, 2015). In addition to carcinogen-induced skin SCC (Liu et al, 2008) and NPC (Yan et al, 2014), IKKα has also been suggested to act as a tumor suppressor in lung small cell carcinoma because kinase dead IKKα knock-in mice, which present only low levels of the IKKα(K44M) mutant protein, exhibit spontaneous lung SCC development and the recruitment of tumor-promoting inflammatory macrophages (Xiao et al, 2013). An in silico meta-analysis of IKKα expression profiles with Kaplan–Meier plotter performed on a cohort of LCs showed that higher expression of IKKα was linked with overall survival in all LC cases and all cases of adenocarcinoma, but not in lung SCC cases (Xiao et al, 2015). Thus, whereas IKKβ-dependent canonical NF-κB activation has been linked to cancer development by multiple studies, IKKα appears to function either as a tumor promoter or a tumor suppressor dependent on the specific cellular context. Recently, two independent teams of investigators have reported contrasting functional roles of lung IKKα in

oncogenic, mutated *K-Ras*–dependent adenocarcinoma development and progression in mice (Song et al, 2018; Vreka et al, 2018). In the case of LC development and progression, IKKα's role could also be dependent on the experimental model used to ablate lung IKKα expression and induce lung adenocarcinomas. Thus, more precisely controlled and cell-targeted experiments using novel in vivo experimental NSCLC models are warranted to better define the mechanisms of action of IKKα to clarify its tumor promoter or suppressor activity in LC development and progression.

Herein, we show that IKKα expressed in lung epithelial cells functions as a major tumor suppressor of NSCLC development and progression in two independent in vivo models: (1) lung adenomas induced by urethane carcinogen in mice with an inducible IKKα deletion targeted solely in AT-II (alveolar type II) lung epithelial cells and (2) a panel of IKKα knockdown human NSCLC lines (harboring either wild-type or mutant *K-Ras* and p53 genes) grown as tumor xenografts in immune-compromised NSG mice. Moreover, we also provide evidence that IKKα's mechanism of action as an NSCLC tumor suppressor can, at least in part, involve the control of hypoxia-inducible pathways required for the enhanced growth of tumors in vivo.

## Results

### IKKα functions as a tumor suppressor in the development of urethane-induced LC in mice

To begin investigating if IKKα in lung epithelial cells has unique effects for the development and progression of NSCLC, we used conditional lung-specific compound transgenic mouse strains in which either IKKα or IKKβ are ablated in an inducible manner only in AT-II lung epithelial cells. For this purpose, we generated lung-specific bitransgenic Sftpc-CreER$^{T2}$:ROSA-fLacz (IKKα$^{f/f}$ and/or IKKβ$^{f/f}$) mouse strains by interbreeding LoxP-targeted IKKα or IKKβ ROSA-fLacz mice (Borzì et al, 2010) and Sftpc-CreER$^{T2}$ mice (Bianchi et al, 2011). Thus, AT-II lung epithelial cells of these two mouse strains express a tamoxifen (Tmx)-inducible Cre recombinase (Sftpc-CreER$^{T2}$) under the transcriptional control of the lung epithelial cell–specific surfactant protein C (SPC) promoter (Xu et al, 2012). 1We induced lung adenoma (AD) development in response to urethane carcinogen administration. Importantly, urethane-induced mouse lung tumors frequently harbor oncogenic *K-Ras* (*K-Ras*4B$^{G12D}$) or EGFR mutations (Kelly-Spratt et al, 2009; de Seranno & Meuwissen, 2010; Diaz et al, 2012; Farago et al, 2012; Misale et al, 2012). In response to urethane, premalignant lung lesions such as focal hyperplasia of type II pneumocytes develop at ~2 wk and progress to adenomas in ~6–8 wk and to malignant adenocarcinomas in ~12–15 wk depending on the mouse strain (Fisher et al, 2001; Ji et al, 2006; Stathopoulos et al, 2007; Kelly-Spratt et al, 2009; de Seranno & Meuwissen, 2010; Xu et al, 2012).

After exposure to a regimen of multiple doses of urethane, 6 mo after the first urethane injection mice lacking IKKα (IKKα$^{KO}$ in AT-II lung epithelial cells), in contrast to wild type (WT) IKKα (IKKα$^{WT}$) control mice, developed a significantly greater number of small (<1 mm size) adenomas and a unique class of much larger tumors (>1 mm in size). Fresh lung specimens fixed in formalin or stained in

H&E are shown in Fig 1A–H The loss of CHUK/IKKα mRNA expression in the urethane-induced large lung adenomas (which solely developed in the Tmx-treated IKKα$^{f/f}$:Rosa-fLacz:Sftpc-CreER$^{T2}$ mice) was confirmed by qRT-PCR analysis of CHUK/IKKα exons 6 and 7, which are flanked by LoxP sites that are specifically deleted by Tmx-induced Sftpc-CreER$^{T2}$–mediated recombination; and the lungs of the same mice as expected are X-gal positive (Fig S1). Scatter plots and statistical analyses comparing the number and sizes of tumors in IKKα$^{WT}$ mice versus mice lacking IKKα in AT-II lung epithelial cells are shown in Fig 1I, clearly revealing that IKKα loss in AT-II lung epithelial cells just before urethane exposure increases the number of small lung adenomas and also results in the appearance of a novel class of much larger lung tumors. In agreement with the IKKα mRNA qRT-PCR results (Fig S1), quantitative Western blotting shows that IKKα protein levels were greatly reduced in the large adenomas isolated from multiple experimental mice (Fig 1J). Importantly, the latter unique, large class of urethane-induced adenomas in the Tmx-treated IKKα$^{f/f}$:Rosa-fLacz:Sftpc-CreER$^{T2}$ mice were not observed amongst the urethane-induced lung adenomas in IKKα$^{WT}$ mice (Fig 1I).

### IKKβ, in contrast to IKKα, acts as a tumor promoter in urethane-induced murine lung adenomas

The induced ablation of IKKβ in AT-II lung epithelial cells significantly reduced the numbers and sizes of lung adenomas 6 mo after the first urethane injection (Fig 2A and B), indicating that IKKβ (unlike IKKα) acts as a promoter of NSCLC development in this in vivo urethane-based LC model. This result is in agreement with earlier work showing that canonical NF-κB signaling and activity is required for LC development induced by urethane (Stathopoulos et al, 2007). Thus, our findings, taken together with these earlier results, indicate that in AT-II lung epithelial cells, IKKβ is likely required to maintain an activated state of canonical NF-κB signaling, which is required for urethane-induced NSCLC development (Stathopoulos et al, 2007; Zaynagetdinov et al, 2012).

### IKKα suppresses tumor growth of several independent human NSCLC cell lines in in vivo murine xenografts

To explore the general significance of lung epithelial IKKα functioning as a murine NSCLC tumor suppressor, we used a second in vivo LC model to investigate the effects of IKKα for the growth of several established human LC lines. We used a panel of three independent human NSCLC lines, which differ either in their p53 or K-Ras functional status (Fig 3A). A novel puromycin-resistant lentiviral vector, which co-expresses a doxycycline-inducible human IKKα shRNA-GFP cassette (Fig 3B), was used to ablate IKKα protein expression in each of the three human NSCLC lines. Control cells were stably transduced with the same lentiviral vector lacking the IKKα shRNA sequence. Western blotting showed efficient IKKα protein knockdowns (IKKα$^{KDs}$) in response to doxycycline in each of the human NSCLC lines (Fig 3C).

When maintained in vivo as tumor xenografts in immune-compromised NSG mice for 3 wk, each of the three IKKα deficient human NSCLC lines presented significantly greater adenocarcinoma

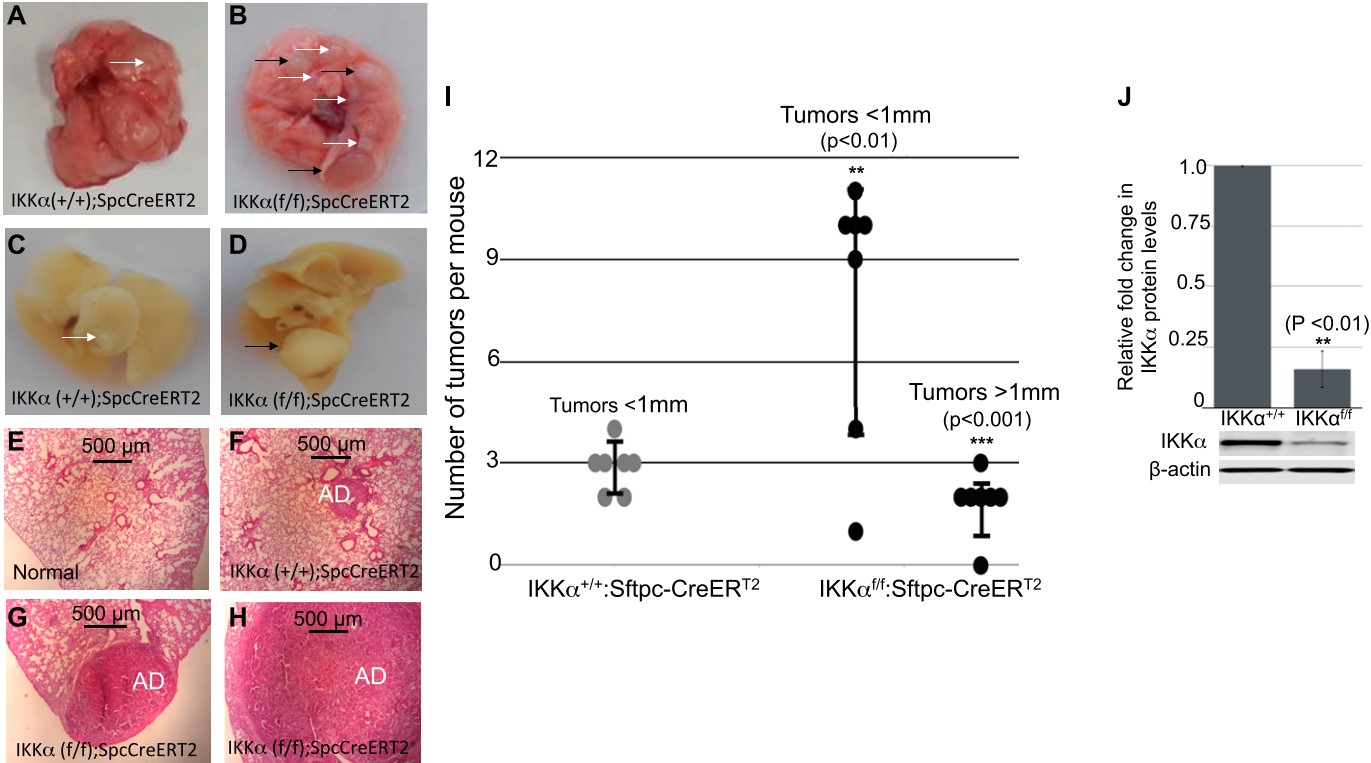

**Figure 1. Conditional deletion of IKKα in AT-II lung epithelial cells increases the number and size of urethane-induced lung adenomas.**
**(A, B, C, D)** Representative photographs of lungs 6 mo after tamoxifen (Tmx) administration and the first urethane injection from control IKKα$^{+/+}$:Sftpc-CreER$^{T2}$ (A, C) and experimental IKKα$^{f/f}$:Sftpc-CreER$^{T2}$ mice (B, D). **(A, B, C, D)** Images of either fresh lung tissues (A, B) or formalin-fixed lung tissues (C, D) are shown with small tumors (<1 mm diameter) indicated by white arrows and large tumors (>1 mm diameter) highlighted by black arrows. **(E, F, G, H)** H&E (hematoxylin-eosin) stains of specific lung tissues are shown as follows: normal lung tissue (E), lung sections of control IKKα$^{+/+}$:Sftpc-CreER$^{T2}$ (F), and experimental IKKα$^{f/f}$:Sftpc-CreER$^{T2}$ (G, H) mice after Tmx and multidose urethane administration. Adenomas (ADs) of different sizes are indicated. As also indicated in Materials and Methods section, we were unable to perform a blinded analysis in the quantification of the mouse tumors because the tumors in the control and experimental mice are clearly of different numbers and sizes; and the unique class of urethane-induced large adenomas in the IKKα$^{KO}$ AT-II lung epithelial cell mice were also visible by eye. **(I)** Scatter plot quantitative analysis of murine lung tumors <1 mm diameter and >1 mm diameter after tamoxifen (Tmx) and multidose urethane administration in control WT IKKα$^{+/+}$:Sftpc-CreER$^{T2}$ and experimental IKKα$^{f/f}$:Sftpc-CreER$^{T2}$ mice (n = 7 mice per group). The number of tumors <1 mm in the control group n = 20 compared with n = 55 for the experimental group. **P < 0.01 by two-tailed t test. Large tumors >1 mm diameter in the control group n = 0 and in the experimental group n = 13. ***P < 0.001 two-tailed t test. Scatter plot ± SD. **(J)** Bar graphs of Western blot results comparing the levels of IKKα versus β-actin proteins in urethane-induced small lung adenomas of IKKα$^{WT}$ control and the unique class of large adenomas in Tmx-treated IKKα$^{f/f}$:Sftpc-CreER$^{T2}$ experimental mice. One representative IKKα and β-actin Western blot is shown adjacent to the bar graphs. **(J)** Relative levels of IKKα protein in (J) were determined by quantitative densitometry scans of the Western blots of multiple independent IKKα$^{WT}$ control (n = 5) and multiple large adenomas of independent experimental IKKα$^{f/f}$:Sftpc-CreER$^{T2}$ mice (n = 5). **P < 0.01 by two-tailed t test. Bars represent means ± SD.

tumor burdens than their IKKα$^{WT}$ controls. Comparisons of tumor xenografts obtained with IKKα$^{WT}$ H1437, A549, and H1299 human NSCLC lines and their IKKα$^{KD}$ derivatives along with statistical analyses clearly show that IKKα protein knockdown in each of these three human NSCLC lines significantly increases their tumor burdens in NSG mice (Fig 4A and B). Importantly, IKKα protein knockdowns in

each of the three human NSCLC lines were also confirmed in their tumor xenografts at the end point of these experiments (Fig 4C). Taken together with the above effects of targeted IKKα deletion in AT-II lung epithelial cells for the development and growth of urethane-induced lung adenomas in mice, our experiments indicate that IKKα acts as an evolutionarily conserved suppressor of

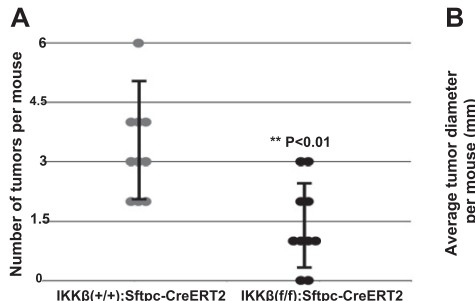

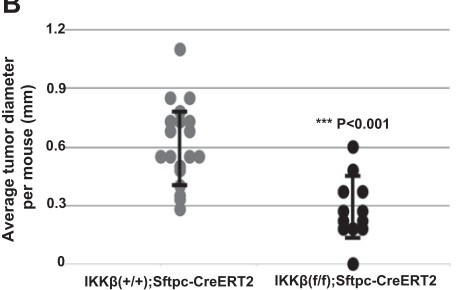

**Figure 2. IKKβ deletion in AT-II epithelial cells reduces the incidence of urethane-induced lung tumors in mice.**
**(A, B)** Quantitative analysis of the numbers (A) and sizes (B) of tumors detected on lung surfaces 6 mo after Tmx administration and the first urethane injection in control IKKβ$^{+/+}$:Sftpc-CreER$^{T2}$ and experimental IKKβ$^{f/f}$:Sftpc-CreER$^{T2}$ mice (n = 10 mice per group). **P < 0.01 and ***P < 0.001 by two-tailed t test. Scatter plots ± SD.

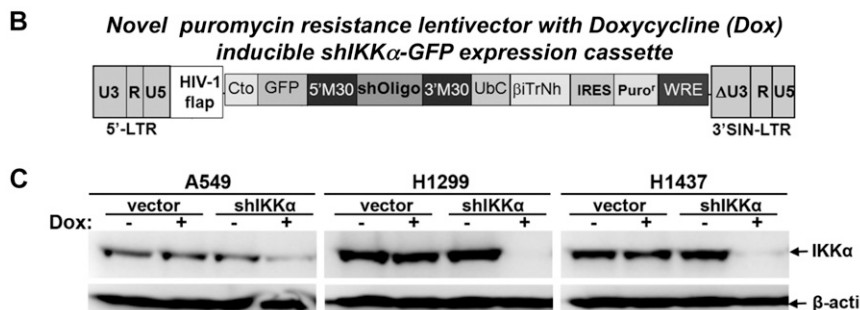

**A**

### Human NSCLC Lines

| CELL LINE | Characteristics | K-Ras | EGFR | p16 | ARF | p53 |
|---|---|---|---|---|---|---|
| A549 | 58 yrs M, Caucasian | K-Ras$^{G12S}$ | wt | -/- (del) | -/- (del) | wt |
| H1299 | 43 yrs M Caucasian | wt | wt | wt | wt | null |
| NCI-H1437 | 60 yrs M Caucasian | wt | wt | mut | mut | p53$^{R267}$ |

**B** *Novel puromycin resistance lentivector with Doxycycline (Dox) inducible shIKKα-GFP expression cassette*

| U3 | R | U5 | HIV-1 flap | Cto | GFP | 5'M30 | shOligo | 3'M30 | UbC | βiTrNh | IRES | Puro$^r$ | WRE | ΔU3 | R | U5 |

5'-LTR    3'SIN-LTR

**C**

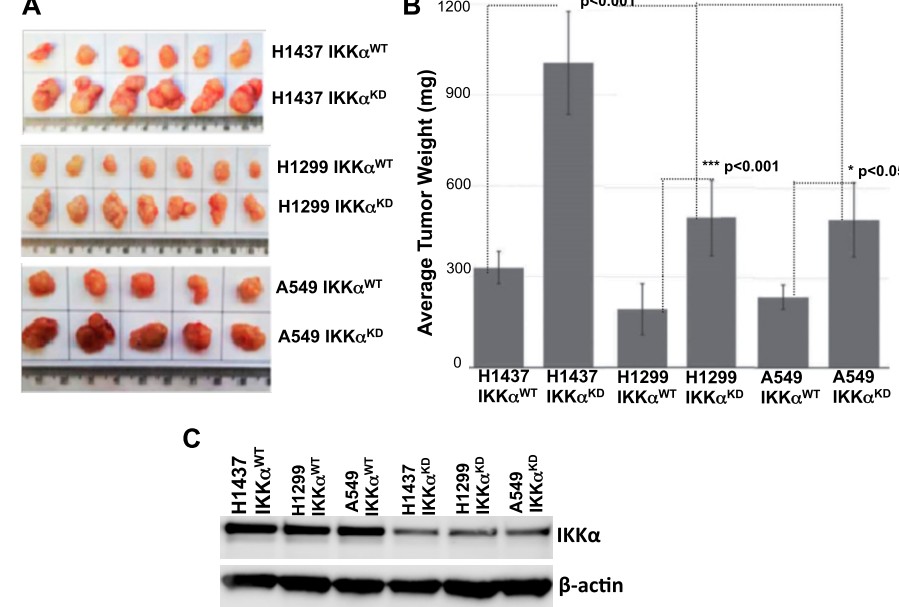

Figure 3.  Generation of IKKα knockdown human NSCLC cell lines.
**(A)** Three different human NSCLC lines were used, which differ in the status of their *K-Ras*, *p53*, *p16*, and *ARF* genes. **(B)** IKKα expression was knocked down (KD) by stable transduction with a puromycin-resistant lentiviral vector that co-expresses a doxycycline (Dox)-inducible human IKKα shRNA-GFP cassette. **(C)** Western blots showing the loss of IKKα protein expression in response to Dox treatment in each of the human NSCLC lines stably transduced with the Dox-inducible IKKα shRNA lentiviral vector, whereas IKKα protein levels remain wild type (WT) in the same NSCLC lines stably transduced with an empty lentiviral vector control lacking the IKKα shRNA insert (IKKα$^{WT}$-lentiviral vector controls). β-Actin protein levels, which are unaffected in all cases, are included as a protein reference control.

murine and human NSCLC growth in vivo. Moreover, because of differences in the status of the *K-Ras* and *p53* genes in these three human NSCLC lines, our results also indicate that IKKα can function as an in vivo NSCLC suppressor in the presence or absence of mutant K-RAS and p53 proteins.

### IKKα functions as an NSCLC suppressor independent of its essential role in the activation of the noncanonical NF-κB p52/RelB pathway

To investigate whether the role of IKKα in lung epithelial cells as an NSCLC tumor suppressor is linked with its essential function as the sole activating kinase of the noncanonical NF-κB/p52-RelB pathway, we knocked down NF-κB p52 expression in A549 and H1299

cells to assess their growth as tumor xenografts in NSG mice compared with their p52$^{WT}$ counterparts. Here, we used a puromycin-resistant retroviral vector that constitutively co-expresses a human p52-targeted shRNA. Representative Western blots showing the p52 protein knockdowns obtained in H1299 and HA549 cultured cells are shown in Fig 5 (panels A and E, respectively). Multiple attempts to generate p52$^{KD}$ H1437 cells were unsuccessful as we were unable to select for viable puromycin-resistant H1437 cells after transduction with the same p52 shRNA retroviral vector, suggesting that reducing NF-κB p52 levels in the H1437 cells compromises their growth in vitro. Thus, we were only able to assess the requirement of the NF-κB p52 subunit for the growth of A549 and H1299 tumor xenografts. Loss of the NF-κB p52 subunit in H1299 and A549 cells significantly reduced their

Figure 4.  H1437, H1299, and A549 human NSCLC cell lines and their IKKα$^{KD}$ derivatives grown in vivo as tumor xenografts.
**(A)** Photographs comparing the sizes of multiple IKKα$^{WT}$ control and IKKα$^{KD}$ tumor xenografts for each of the 3× human NSCLC lines. **(B)** Quantitative analysis comparing the tumor burdens for each of the IKKα$^{WT}$ and IKKα$^{KD}$ human NSCLC lines. **(C)** Representative Western blots confirming the loss of IKKα protein expression in IKKα$^{KD}$ H1437, A549, and H1299 xenografts compared with each of their IKKα$^{WT}$ control xenografts. IKKα knockdown (KD) in each cell line was induced by exposure to Dox before injection into NSG mice. Each cell line was maintained for 3 wk in NSG mice (2 × 10$^6$ cells per injection), and IKKα$^{KD}$ in vivo was maintained by adding Dox to the mouse drinking water. Importantly, each human NSCLC line IKKα$^{WT}$ positive control was stably transduced with an empty version of the same lentiviral vector lacking the IKKα shRNA sequence, and Dox was also added to the drinking water of these control mice. As shown in (B), significant differences were observed in tumor burdens comparing vector controls with their IKKα$^{KD}$ derivatives for each of the three human NSCLC lines 3 wk after their inoculation into NSG immune-compromised mice. H1437 (n = 6) ***P < 0.001; H1299 (n = 7) ***P < 0.001; A549 (n = 5) *P < 0.05 by two-tailed t tests. Moreover, H1437 IKKα$^{KD}$ tumor burdens are more robust than the ones observed for A549 IKKα$^{KD}$ or H1299 IKKα$^{KD}$ (***P < 0.001 by one way ANOVA t test). Bars represent means ± SD.

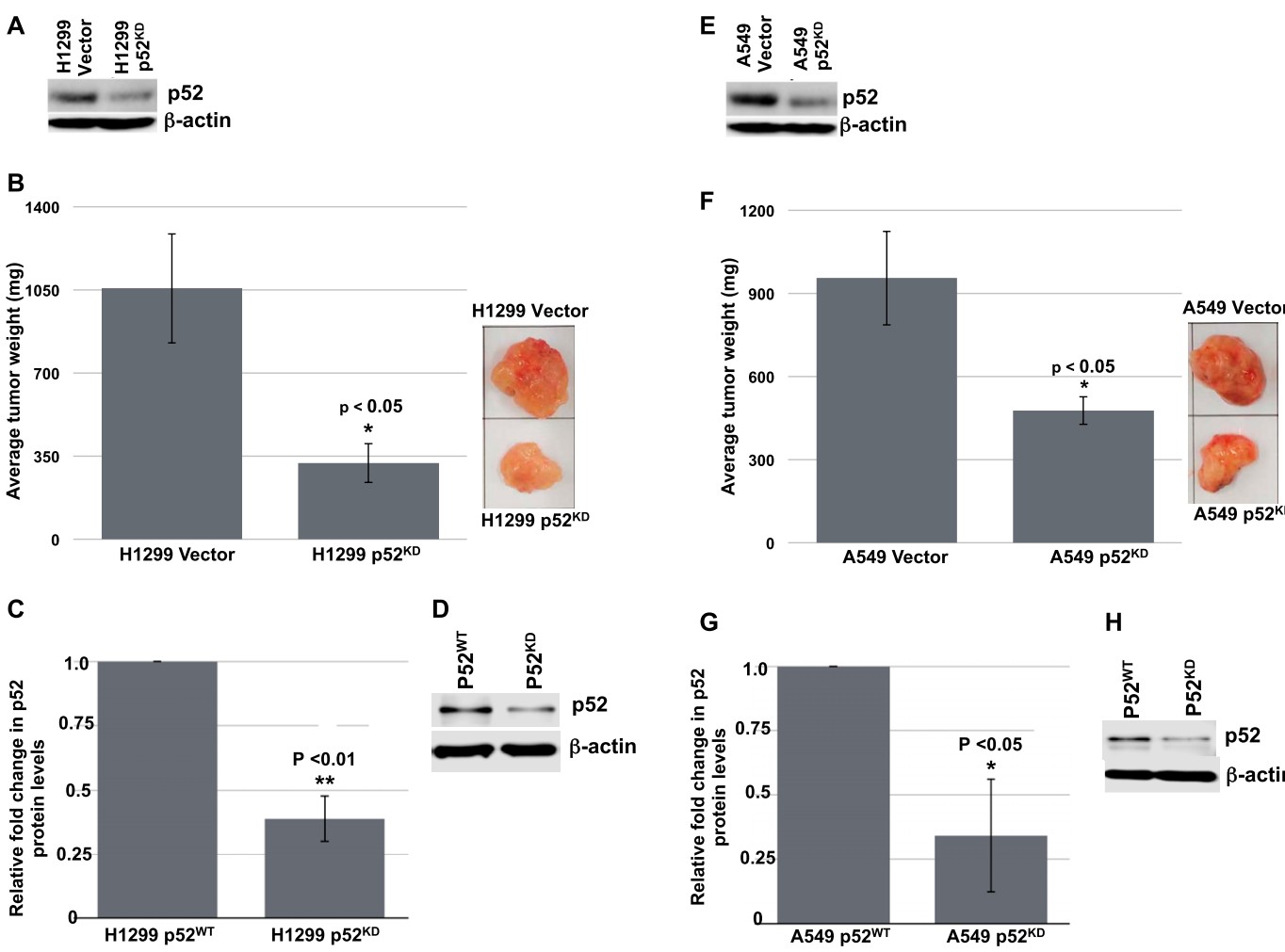

**Figure 5. Knockdown of the NF-κB p52 subunit reduces the tumor burdens of H1299 and A549 tumor xenografts in immune-compromised NSG mice.**
**(A)** Western blot showing the knockdown of the p52 subunit in cultured H1299 cells stably transduced with a Moloney retroviral vector that co-expresses a puromycin-resistant gene and a human NF-κB p52 shRNA versus H1299 vector control cells, which were stably transduced with the same retroviral vector lacking the p52 shRNA sequence. **(B)** Bar graphs comparing the tumor burdens of multiple WT and p52^KD H1299 tumor xenografts alongside representative tumor photographs. **(C)** Bar graphs showing comparisons of p52 protein levels obtained from densitometry scans of p52 and β-actin Western blots of multiple independent WT and p52^KD H1299 tumor xenografts. **(C, D)** Photograph of a representative Western blot used for the bar graph analysis in (C). **(E)** Western blot showing the knockdown of the p52 subunit in cultured A549 cells compared with their p52^WT control. **(F)** Bar graphs comparing the tumor burdens of multiple WT and p52^KD A549 tumor xenografts alongside representative tumor photographs. **(G)** Bar graphs of multiple Western blots confirming the loss of p52 protein expression in p52^KD A549 tumor xenografts. **(G, H)** Photograph of a representative Western blot used for the bar graph analysis in (G). Relative tumor burdens and p52 protein levels were quantified (H1299 n = 5 and A549 n = 6). *$P < 0.05$ and **$P < 0.01$ by two-tailed $t$ test. Bars represent means ± SD.

tumor xenograft burdens in NSG mice as shown in Fig 5 (panels B and F, respectively). Importantly, p52 protein knockdowns in the relevant tumor xenografts at the end point of these experiments were also verified by immunoblotting (H1299 in Fig 5C and D and A549 in Fig 5G and H). These results indicate that IKKα acts as an NSCLC tumor suppressor independent of its unique function to activate the p52-dependent noncanonical NF-κB pathway.

**HIF-1α protein and its direct target genes are up-regulated in IKKα^KO urethane-induced large lung adenomas and in IKKα^KD H1437 tumor cells grown in vivo as tumor xenografts or maintained in vitro under hypoxic conditions**

To begin to explore the possible mechanisms of action of IKKα in murine and human lung epithelial cells as an NSCLC tumor

suppressor, we performed a series of whole transcriptome sequencing experiments followed by bioinformatics analysis. Multiple RNA-seq experiments were carried out to compare the transcriptomes of: (1) urethane-induced small adenomas in IKKα^WT mice versus the unique class of large adenomas in AT-II lung epithelial IKKα^KO mice; (2) human H1437, A549, and H1299 NSCLCs with or without IKKα knockdown grown as tumor xenografts in NSG immune-compromised mice. Heat maps of selected differentially expressed genes (DEGs) in the IKKα^KO urethane-induced large mouse adenomas and the tumor xenografts of the IKKα^KD human NSCLC cells reveal a paucity of genes with similar up- or down-modulations (DEG selection criteria is described in the Materials and Methods section) (Fig S2). However, a subset of DEGs were found to be similarly modulated in murine urethane-induced NSCLCs and the H1437 IKKα^KD tumor xenografts (Fig S2). To

determine if specific pathways associated with tumor develop-ment and growth could be altered in response to the IKKα loss, we used Ingenuity upstream regulator analysis, which infers up-stream regulator activation based on differentially expressed transcripts using Ingenuity's manually curated knowledge base (QIAGEN Inc., https://www.qiagenbio-informatics.com/products/ingenuity-pathway-analysis). As shown by heat maps in Fig 6, In-genuity analysis predicted the differential activation of specific HIF-1α direct target genes in the mouse urethane-induced large IKKα$^{KO}$ lung adenomas (Fig 6A) and in the IKKα$^{KD}$ H1437 tumor xenografts (Fig 6B). Interestingly, the one up-regulated DEG that is shared in common between the murine IKKα$^{KO}$ large lung adenomas and xenografts of the H1437, A549, and H1299 IKKα$^{KD}$ human NSCLCs is HIGD2A (hypoxia-inducible domain family member 2A) encoding a cytochrome c oxi-dase complex (complex IV) subunit; HIGD2A is the terminal enzyme in the mitochondrial respiratory chain that is regulated by HIF-1α (Ameri et al, 2013) and has been shown (akin to the related protein HIGD1A) to enhance cell survival under hypoxia (An et al, 2011; Salazar et al, 2019).

To experimentally validate the Ingenuity bioinformatics analysis, we investigated the expression levels of HIF-1α protein and se-lected direct HIF-1α target genes. In the urethane-induced NSCLC mouse model, we compared the levels of HIF-1α protein in the small lung adenomas of multiple IKKα WT control IKKα$^{+/+}$:Sftpc-CreER$^{T2}$ mice with the amounts of HIF-1α expressed by the unique class of large adenomas in the experimental IKKα$^{f/f}$:Stpc-CreER$^{T2}$ mice (Fig 7). Indeed as shown by either quantitative immunblotting (Fig 7A and B) or immunofluorescence (IF) analysis (Fig 7C), each of 6 in-dependent IKKα$^{KO}$ large lung adenomas have significantly en-hanced HIF-1α protein levels. In light of these observations, we next investigated if the higher levels of HIF-1α protein in the lungs and the large adenomas of the urethane-treated experimental IKKα$^{f/f}$:Sftpc-CreER$^{T2}$ mice also resulted in the up-regulation of specific direct HIF target genes. We used qRT-PCR to quantify the expression levels of the hexokinase 2 (*HK2*) and glucose transporter (*Slc2a1/Glut1*) genes (results shown in Fig 8A and B, respectively), which are well-known direct HIF targets, required for the initiation of gly-colysis and glucose transport. These two glycolytic enzymes have previously been found to be up-regulated in a variety of malignant cancers including LC (Iyer et al, 1998; Amann et al, 2009; Menendez et al, 2015; Ooi & Gomperts, 2015). Indeed, the *HK2* and *Scl2a1/Glut1* genes were both significantly up-regulated in each IKKα$^{KO}$ urethane-induced large adenoma as compared with multiple IKKα$^{WT}$ controls (Fig 8A and B).

Next, we explored the activation status of HIF-1α in H1437 IKKα$^{KD}$ cells grown as tumor xenografts in immune-compromised NSG mice or maintained in vitro under hypoxic conditions either as monolayer cells or tumor spheres. Quantitative Western blotting revealed that H1437 IKKα$^{KD}$ xenografts express significantly higher levels of HIF-1α protein than their IKKα$^{WT}$ controls (Fig 9A). Im-portantly, quantified Western blotting in Fig 9B showed greatly reduced levels of IKKα protein in the same H1437 IKKα$^{KD}$ xenografts (in accord with the H1437 IKKα protein blots in Figs 3C and 4C). To determine if the H1437 IKKα$^{KD}$ cells had similar properties under hypoxic conditions in vitro, we grew them as monolayer cells or tumor spheres under normal conditions (normoxia) or under 3% hypoxia. Importantly, we initially observed that IKKα$^{WT}$ and IKKα$^{KD}$

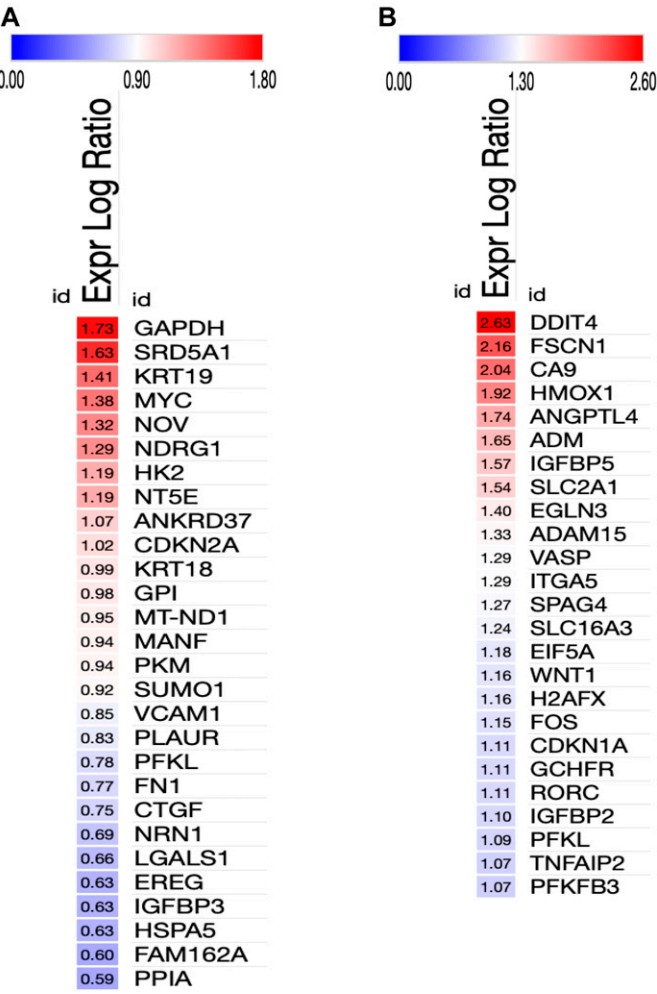

Figure 6. Heat maps displayed from ingenuity bioinformatics analysis predict the differential activation of specific HIF-1α direct target genes.
**(A)** IKKα$^{KO}$ versus IKKα$^{WT}$ urethane-induced mouse lung tumors. **(B)** IKKα$^{KD}$ versus IKKα$^{WT}$ H1437 tumor xenografts. Relative fold change values were uploaded onto the Morpheus analysis tool (https://software.broadinstitute.org/morpheus). Red and blue colors denote fold change increases or decreases in the expression of specific HIF-1α target genes in the IKKα$^{KO}$ (A) and IKKα$^{KD}$ (B) tumors versus their respective IKKα$^{WT}$ controls with numerical values denoting fold changes in gene expression.

H1437 tumor spheres grew similarly under normoxia; but IKKα$^{KD}$ H1437 tumor spheres grew better under hypoxic conditions than IKKα$^{WT}$ H1437 cells (Fig S3A and B), indicating that IKKα$^{KD}$ enhanced the tumor cell adaptability to hypoxia. In agreement with these results, quantitative Western blotting revealed that HIF-1α protein was significantly up-regulated in IKKα$^{KD}$ H1437 cells grown under 3% hypoxic conditions (Fig 9C), and the IKKα protein knockdowns were also validated in the same cells (see representative Western blot in Fig 9D). To confirm our HIF-1α protein results by another method, quantitative IF analysis was performed and also revealed signifi-cantly higher levels of HIF-1α protein in H1437 IKKα$^{KD}$ cells grown under hypoxic conditions in vitro as either monolayer cells (Fig 10A and B) or tumor spheres (Fig 10C and D). Finally, as observed with the mouse IKKα$^{KO}$ large adenomas, two selected direct HIF-1α target genes (Slc2a1 and PDK1) were also found to be up-regulated by

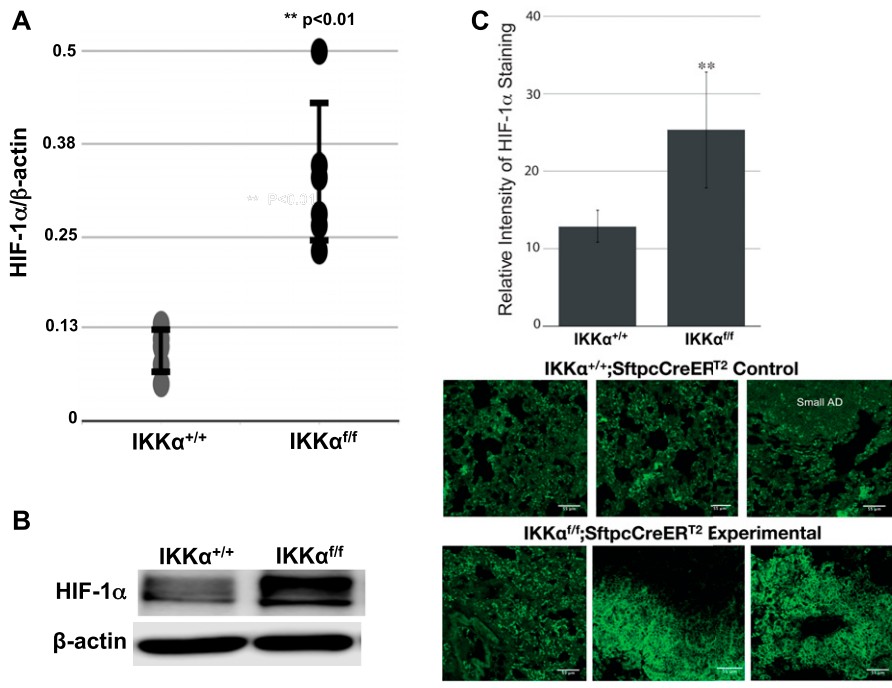

**Figure 7.  Quantitative comparisons of HIF-1α protein levels in lung tumors/tissues of IKKα$^{+/+}$ versus IKKα$^{f/f}$:Sftpc-CreER$^{T2}$ mice after their exposure to tamoxifen and multidose urethane regimen over a 6-mo time course.**
**(A)** Scatter plots of the relative levels of HIF-1α protein obtained from densitometry scans of HIF-1α and β-actin Western blots of 6× IKKα$^{WT}$ small lung adenomas and 6× IKKα$^{KO}$ large lung adenomas all from independent mice (**$P < 0.01$ by two-tailed $t$ test; scatter plot ± SD). **(B)** Photograph of one representative immunoblot showing enhanced HIF-1α protein expression in an IKKα$^{KO}$ large lung adenoma. **(C)** Bar graph results of IF analysis quantifying the HIF-1α protein expression in the lung tissues of the same mice. Mean fluorescence intensities of HIF-1α protein in single optical sections of images were acquired with confocal microscopy and quantified by ImageJ (**$P < 0.01$ by two-tailed $t$ test). Bars represent means ± SD. For IF analysis, paraffin sections (5 μm) of mouse lung tissues were deparaffinized and submitted to IF by incubating with a primary anti-HIF-1α antibody followed by a secondary GFP-conjugated antibody as described in Materials and Methods section. **(C)** Representative IF images of HIF-1α protein expressed by IKKα$^{WT}$ control and conditional IKKα$^{KO}$ lung sections from multiple independent mice are shown just below the bar graph analysis (C). Cytoplasmic and nuclear HIF-1α protein staining was apparent in the IF images of the experimental IKKα$^{f/f}$:Sftpc-CreER$^{T2}$ mouse lungs. Scale bar = 55 μm.

quantitative qRT-PCR in independent IKKα$^{KD}$ H1437 tumor xenografts (Fig S4A and B). Taken together, our findings in two independent in vivo models indicate that IKKα in lung epithelial cells functions as a tumor suppressor by reducing murine NSCLC growth and human NSCLC tumor xenograft burden, at least in part, by inhibiting/repressing HIF-1α protein accumulation and the expression of HIF direct target genes.

## Discussion

Herein, we have used murine and human in vivo NSCLC models definitively, showing that lung epithelial IKKα functions as an evolutionarily conserved NSCLC tumor suppressor. In urethane, carcinogen-induced murine lung adenomas that develop over a 6-mo time course, the induced ablation of IKKα solely in lung AT-II lung epithelial cells increased both the number and size of lung adenomas. Moreover, mice lacking IKKα in their lung AT-II epithelial cells, when exposed to urethane, presented 1–2 large (>1 mm diameter) adenomas in their lungs, which were not at all observed in IKKα$^{WT}$ mice exposed to the same urethane regimen. We hypothesize that the loss of IKKα is linked to more than one rare urethane-induced genetic or epigenetic alteration, initially resulting in an increased number of small adenomas, one or two of which eventually evolve into much larger lung tumors in each animal. Unlike IKKα, the loss IKKβ in AT-II lung epithelial cells in mice led to reduced tumor development in response to urethane exposure. This latter observation is in agreement with earlier work from several groups which showed that canonical NF-κB signaling and activity is required for LC development induced by urethane (Stathopoulos et al, 2007; Meylan et al, 2009; Basseres et al, 2010; Xia et al, 2012). In accord with these earlier reports, we conclude that the induced ablation of IKKβ in lung AT-II epithelial cells in our urethane-induced NSCLC mouse model blocks the activation of canonical NF-κB signaling in the airway epithelium, which is required for urethane-induced NSCLC development (Stathopoulos et al, 2007; Zaynagetdinov et al, 2012).

To investigate if IKKα also functions as a human NSCLC tumor suppressor, we knocked down IKKα expression in a panel of three well-characterized, malignant human NSCLC lines (H1437, A549, and H1299). Indeed, the loss of IKKα in these unrelated human NSCLC cell lines enhanced their growth as tumor xenografts in immune-compromised NSG mice, indicating that IKKα loss also rapidly collaborates with pre-existing genetic and/or epigenetic alterations in malignant human NSCLCs to make them more aggressively growing cancers. As H1437, A549, and H1299 have either activated, oncogenically mutated K-Ras (A549) or WT K-Ras genes (H1437 and H1299), IKKα can function as an NSCLC tumor suppressor with or without oncogenic K-Ras. Moreover, the different status of the p53 tumor suppressor genes in these three NSCLC lines (A549 WT p53, H1299 p53-null, and H1437 p53 gain of function oncogenic mutation) also shows that IKKα's NSCLC tumor suppressor activity can be independent of p53 status. In addition, and in contrast to IKKα loss, knockdown of the noncanonical NF-κB p52 subunit in the H1299 and A549 NSCLC lines reduced their tumor xenograft burdens in immune-compromised NSG mice, indicating that IKKα's NSCLC tumor suppressor activity is independent of its unique function to activate the NF-κB noncanonical signaling pathway. In agreement with our results, NF-κB p52 has been previously shown to enhance LC progression (Giopanou et al, 2015; Saxon et al, 2018). Thus, taken together, our results indicate that IKKα in lung epithelial cells can

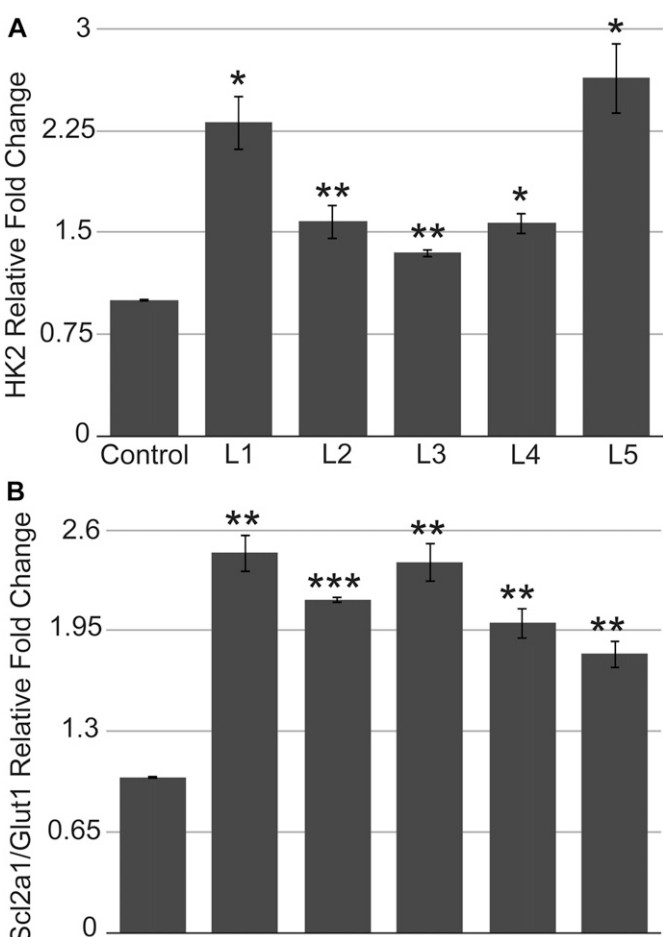

**Figure 8. Expression of direct HIF target genes.**
**(A, B)** qRT-PCR results showing up-regulation of direct HIF-1α target genes HK2 (A) and Scl2a1/Glut1 (B) in each of five independent IKKα[KO] urethane-induced large adenomas (L1–L5) compared with three independent IKKα[WT] control urethane-induced small lung adenomas. **P < 0.05, **P < 0.01, ***P < 0.001 by two-tailed t test. Bars represent means ± SD.

function as an evolutionarily conserved NSCLC tumor suppressor independent of oncogenic K-Ras, p53, and the noncanonical NF-κB pathway. IKKα's ability to function as an NSCLC tumor suppressor in lung epithelial cells is most likely because of its other properties, including NF-κB–independent nuclear functions, a number of which have been previously described, including: (a) IKKα has an NLS (Connelly & Marcu, 1995) and was observed by IF analysis in the nuclei of all three human NSCLC lines used in our study (data not shown); (b) IKKα is known to regulate the expression of genes as a chromatin modifier independent of NF-κB (reviewed in Scheidereit (2006); Perkins (2007); Perkins (2012)); (c) IKKα in association with specific SMAD TFs positively regulates the expression of c-Myc antagonists in the context of epidermal cell differentiation (Descargues et al, 2008); and (d) independent of its kinase activity, IKKα has also been shown to positively regulate the expression of MMP-10, an activator of MMP-13 enzymatic activity, which is required for chondrocytes to differentiate towards a hypertrophic phenotype (Olivotto et al, 2013).

## Contrasting views on functional roles of IKKα in LC development

When we were completing our experiments and in the process of preparing our article, two groups reported contrasting views on the functional role of IKKα in K-Ras–dependent NSCLC development. Vreka and colleagues reported that IKKα acts as a tumor promoter in K-Ras mutant NSCLC (Vreka et al, 2018), and just before their study, a published report by another group showed that IKKα instead acts as a tumor suppressor in oncogenic K-Ras–induced NSCLC (Song et al, 2018). In both studies, the IKKα-encoding gene was deleted by Cre-mediated Lox P recombination induced by intratracheal adenovirus Cre (Ad Cre) administration, which would infect all cell types in lung airways including ciliated cells, secretory cells, Clara cells, basal cells, and other undifferentiated cells (Mastrangeli et al, 1993). In addition, Vreka and colleagues also used another approach to delete IKKα in lung epithelial cells by using an SPC-Cre transgene, a strategy that would delete IKKα in SPC-positive cells during early embryonic development, which was previously reported to result in the formation of abnormal dilated cysts likely because of a Cre-specific toxicity leading to excessive apoptosis (Jeannotte et al, 2011). Nevertheless, in spite of these significant issues, the reasons for the contrasting findings of these two earlier reports remain unknown, and thus, it remains rather important to continue to investigate IKKα's in vivo function in the development and progression of NSCLC using other well-controlled targeted models of genetic perturbation.

In our murine urethane-induced NSCLC in vivo model, we specifically induced IKKα deletion only in adult mouse AT-II lung epithelial cells just before urethane administration. Moreover, the report by Song et al (2018), which reached the same conclusion as our work (i.e., IKKα functions as an NSCLC tumor suppressor), only induced NSCLC development in response to oncogenically activated K-Ras (Song et al, 2018). This very specific oncogenic K-Ras–dependent in vivo model revealed that IKKα loss enhanced reactive oxygen species (ROS) activity by up-regulating Cybb/NOX2 (a subunit of the NADPH/NOX oxidase complex involved in ROS generation) and also by suppressing the ROS negative feedback regulator NRF2 (Song et al, 2018). In contrast, another group using conditional KRas[G12D] oncogene expression in C57BL/6J mice reported that NRF2 activation prevents initiation of chemically induced cancer, although it promotes progression of pre-existing tumors regardless of any chemical or genetic etiology (Tao et al, 2014). Moreover, after initiation of lung tumors, NRF2 inhibition can suppress the progression of chemically and spontaneously induced tumors (Tao et al, 2014). It is also important to note here that other groups have reported an increase in HIF responses upon NRF2 induction (Malec et al, 2010) and that HIF and NRF2 can also act synergistically to enhance the ability of tumors to grow in hypoxia (Toth & Warfel, 2017), which would be in accord with some of the observations reported by Tao et al (2014). Importantly, our experiments herein were not solely focused on NSCLC induced by activating oncogenic K-Ras mutations. In addition, we also observe that NRF2/NFE2L2 was amongst the up-regulated DEGs in urethane-induced IKKα[KO] large lung tumors and was also somewhat up-regulated in K-Ras mutant A549 IKKα[KD] tumor xenografts (see heat maps in Fig S2). However, it was down-regulated in IKKα[KD] H1437 xenografts and essentially unaffected in IKKα[KD] H1299 xenografts,

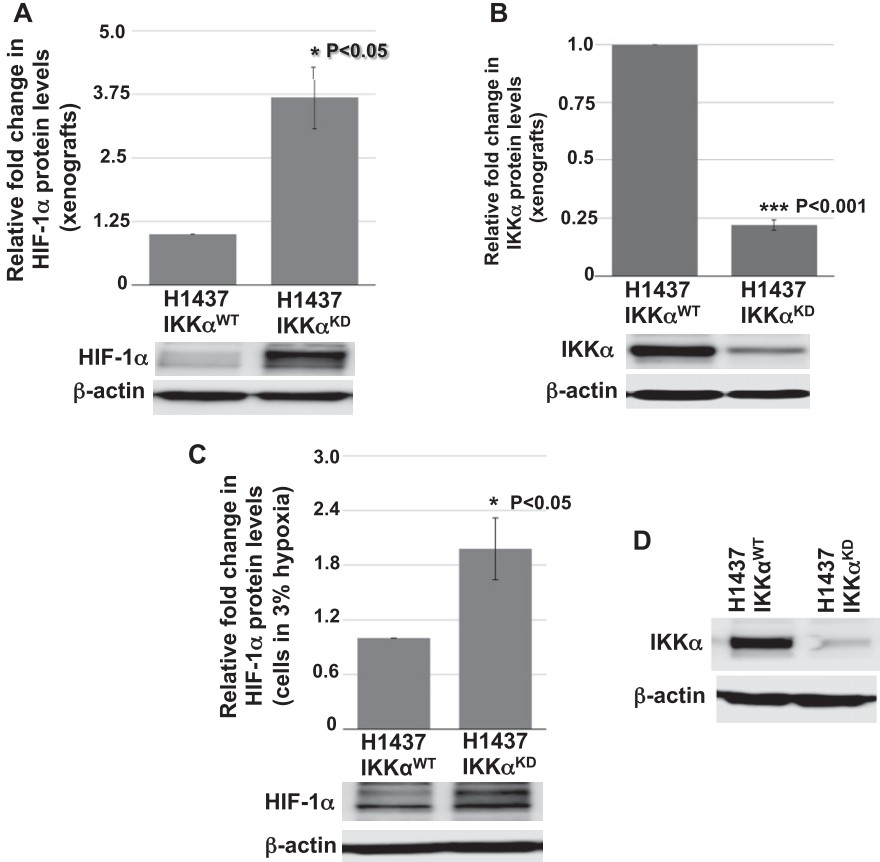

**Figure 9. Immunoblot quantifications of HIF-1α protein expression in IKKα^WT and IKKα^KD H1437 tumor xenografts and monolayer cells exposed to 3% hypoxia in vitro.**
**(A)** Bar graph analysis of the relative levels of HIF-1α protein obtained from quantitative densitometry scans of HIF-1α Western blots of total cellular proteins isolated from three independent (same number of the ones used in the RNA-seq analysis) IKKα^WT and IKKα^KD H1437 tumor xenografts and an image of one representative immunoblot. **(A, B)** Quantitative bar graph analysis of Western blots of the relative levels of IKKα protein in the same xenografts from (A) and an image of one representative immunoblot. **(C)** Quantitative bar graph analysis of HIF-1α protein in multiple Western blots of total cellular proteins isolated from IKKα^WT and IKKα^KD H1437 cells grown under 3% hypoxic conditions for 5 h in vitro. **(C, D)** One representative HIF-1α and β-actin immunoblot used for the quantitative analysis in (C). Expression levels of β-actin protein were used as reference controls in all cases. **P < 0.05 and ***P < 0.001 by two-tailed t test. Bars represent means ± SD.

which are both wild type *K-Ras* NSCLCs (Fig S2). Thus, taking together all prior published work and our results herein, it is reasonable to conclude that the nature of the in vivo LC model (including how, where and when IKKα is deleted and also how the LCs are induced or what other oncogenic mutations they may possess) can have pronounced effects on experimental outcomes for IKKα's role in NSCLC development and growth. In this context, our findings herein have revealed that IKKα functions as a potent tumor growth suppressor in both mouse and human NSCLC in vivo models targeted to murine lung epithelial cells or human epithelial NSCLCs, with or without activating, oncogenic *K-Ras* or wild-type *p53* tumor suppressor genes. Moreover, we find that lung epithelial IKKα functions as a tumor suppressor by reducing both murine NSCLC growth and human NSCLC tumor xenograft burden, at least in part, by inhibiting/repressing HIF-1α protein accumulation and its direct target genes.

### IKKα's mechanism of action as a novel NSCLC tumor suppressor is associated with the regulation of the ability of cancer cells to grow in hypoxic environments

Our findings provide evidence for the first time that IKKα's mechanism of action as an NSCLC tumor suppressor in lung epithelial cells can be associated with its ability to function as a negative upstream regulator of HIF-1α protein activity. Ingenuity upstream analysis predicted the activation of the HIF-1α upstream regulator based on the enrichment of differentially expressed transcripts in our transcriptomic analysis, which overlap with Ingenuity's knowledge base for HIF-1α downstream targets. As this HIF-1α target gene signature is manually curated from a variety of independent studies using different cell types and tissues, it is composed of a mixture of HIF-1α signatures. Moreover, because several studies have indicated that HIF-1α–induced genes may differ in different cell types and cancer cells with diverse mutations (Benita et al, 2009; Dengler et al, 2014), HIF-1α activation may have led to the activation of different downstream target genes in our two independent NSCLC in vivo models. In our experiments, enhanced HIF-1α protein expression and activity in the absence of IKKα was most apparent in the murine urethane-induced NSCLC in vivo model and in human H1437 IKKα^KD NSCLC cells grown as tumor xenografts or maintained in vitro either as tumor spheres or monolayer cells under hypoxic conditions. Moreover, activation of HIF-1α in the H1437 IKKα^KD cells was also correlated with their more robust tumor burden as tumor xenografts compared with IKKα^KD H1299 and A549 xenografts. However, there were few HIF-1α target genes shared amongst the up-regulated DEGs in the absence of IKKα in our mouse urethane NSCLC model versus three unrelated human NSCLCs (H1437, A549, and H1299) grown in vivo as tumor xenografts, in agreement with previously published findings (Benita et al, 2009; Dengler et al, 2014), and thus suggesting that there is potential variability in the HIF-1α target gene signatures of unrelated NSCLC tumors with different activating mutations. To

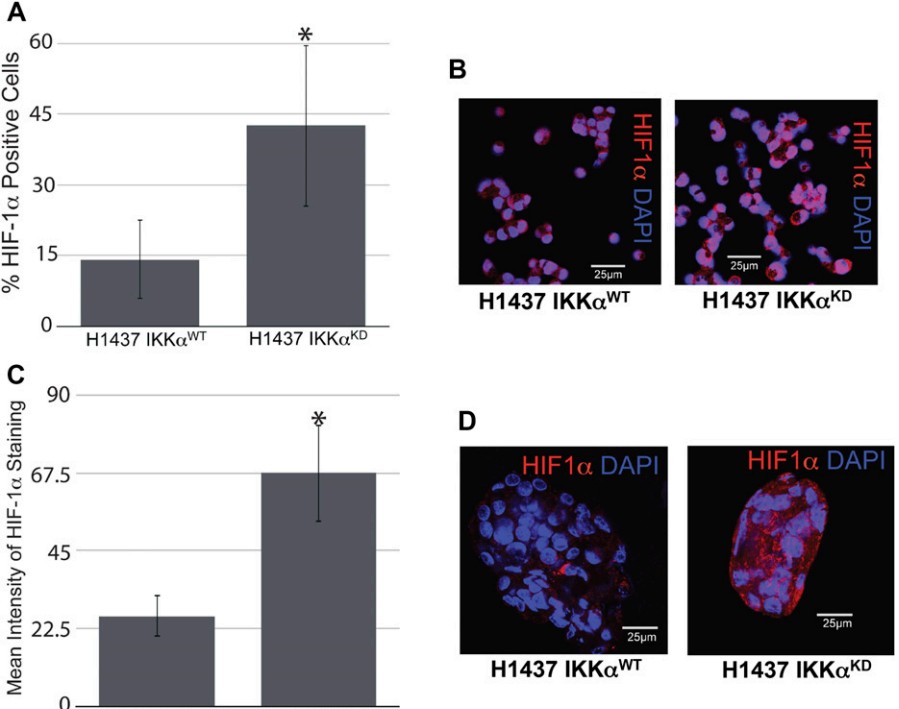

**Figure 10. HIF-1α protein expression revealed by quantitative IF analysis of IKKα^WT and IKKα^KD H1437 exposed as monolayers to 3% hypoxia or grown as tumor spheres under 3% hypoxia.**
**(A)** Quantitative bar graph analysis of the percentages (%) of HIF-1α–positive cells revealed by IF analysis of monolayer cells under 3% hypoxia for 5 h. **(B)** Representative images of the monolayer cells (exposed to anti–HIF-1α and DAPI counterstained), which were used for the quantitative analysis in (A). **(C)** Bar graph analysis of the mean HIF-1α protein intensities obtained by IF analysis of tumor spheres grown under 3% hypoxia for 7 d. **(D)** Representative images of tumor spheres (exposed to anti–HIF-1α and DAPI counterstained), which were used for the quantitative analysis in (C). *$P < 0.05$ by two-tailed $t$ test. Bars represent means ± SD.

validate the role of HIF-1α in our two NSCLC models, we performed experiments to assess HIF-1α activity using known downstream HIF-1α direct target genes. Indeed, SLC2A1 (which was up-regulated in our RNA-seq analysis as shown in Fig S2A and B) and HK2 (which was not amongst the DEGs in Fig S2) were both found to be significantly up-regulated in qRT-PCR analysis of the mouse IKKα^KO urethane-induced large lung adenomas (Fig 8A and B); and SLC2A1 was also found to be enhanced in the IKKα^KD H1437 tumor xenografts (Fig S4A). This discrepancy in HK2 expression may be due to differences in the techniques used to assess the expression of this transcript (e.g., RNA-seq versus qRT-PCR) or the depth of coverage in our RNA-seq results. Overall, we believe that, taken together, our quantitative HIF-1α protein and qRT-PCR analysis of selected direct HIF-1α target genes suggests that IKKα loss can promote the activation of HIF-1α, which enhances NSCLC growth under hypoxic conditions in vivo.

In our studies, there was a pronounced effect of IKKα loss on enhancing HIF activity in mouse relative to human tumors. This pronounced effect of IKKα loss on enhancing HIF-1α activity in the mouse urethane NSCLC model may, in part, be explained by the induction of IKKα KO in lung epithelial cells before urethane exposure. In H1437, A549, and H1299 cells, IKKα was knocked down in established, fully malignant NSCLCs, which had already adapted to grow under in vivo hypoxic conditions. Indeed, there was no apparent induction of HIF-1α pathways in IKKα^KD A549 and H1299 tumor xenografts, even though IKKα knockdown also enhanced their tumor burdens (albeit to a lesser degree than observed for IKKα^KD H1437 cells in which IKKα loss resulted in the up-regulation of HIF-1α protein levels and activity). As mentioned above, it is

important to note in this context that IKKα knockdown in *K-Ras* mutant A549 cells was previously reported to result in ROS up-regulation, which was also shown to contribute to lung tumor growth in a lung-targeted *K-Ras* mutant mouse model (Song et al, 2018). Therefore, our results, taken together with another previously published report, indicate that IKKα in lung epithelial cells can function as an evolutionarily conserved NSCLC tumor suppressor by multiple independent mechanisms (ROS up-regulation and/or HIF-1α up-regulation and activation), suggesting that the mechanisms whereby IKKα functions as an NSCLC suppressor may depend on the nature of the oncogenic mutations responsible for the development and growth of specific NSCLCs. Importantly, as the genes encoding murine and human HIF-1α were not found to be up-regulated in the absence of IKKα in our murine and human NSCLC in vivo models, IKKα would then appear to act as an NSCLC tumor suppressor in an indirect manner by suppressing the accumulation and/or stability of HIF-1α protein under hypoxic conditions.

Interestingly, the HIF-1α gene has previously been shown to be a direct, induced target of IKKβ-dependent canonical NF-κB signaling (Rius et al, 2008; Culver et al, 2010; D'Ignazio et al, 2017). Thus, these earlier findings with IKKβ, taken together with our IKKα results, reveal that IKKβ and CHUK/IKKα have opposing functions for the induction of HIF-1α–dependent pathways, which are required for the enhanced growth of solid tumors under in vivo hypoxic conditions. Future work will be required to determine the mechanism whereby the loss of IKKα can result in greater HIF-1α protein accumulation and activity to facilitate enhanced lung tumor growth under hypoxic conditions.

# Materials and Methods

### IKKα and IKKβ conditional KO mice

Mice with IKKα or IKKβ alleles containing LoxP recombination sites (Penzo et al, 2010) and a ROSA-fLacz Cre-inducible LacZ reporter gene (Soriano, 1999) were crossed with Sftpc-CreER$^{T2}$ mice (Rock et al, 2011), which harbor a tamoxifen-inducible CreER$^{T2}$ recombinase under the control of the Sftpc gene promoter that is only active in AT-II lung epithelial cells. Breeding to generate these bitransgenic IKKα$^{f/f}$:ROSA-fLacz:Sftpc-CreER$^{T2}$ and IKKβ$^{f/f}$:ROSA-fLacz:Sftpc-CreER$^{T2}$ mouse strains was approved by Stony Brook University's IACUC committee in accordance with the USA NIH grant guidelines. All NSCLC experiments with these mice were performed in the animal house facility of the Biomedical Research Foundation Academy of Athens (Athens, Greece), where mice were housed in individually ventilated cages under specific pathogen-free conditions, in full compliance with Federation of Laboratory Animal Science Associations; all procedures for the care and treatment of animals were approved by the Institutional Committee on Ethics of Animal Experiments and the Greek Ministry of Agriculture. To induce either IKKα or IKKβ deletion in AT-II lung epithelial cells, 6-wk-old male or female mice were injected i.p. for five consecutive days with 2 mg tamoxifen (Sigma-Aldrich) dissolved in corn oil. To induce NSCLCs, 1 wk after tamoxifen administration, mice received weekly i.p. urethane injections (1 g/kg) (Sigma-Aldrich) dissolved in PBS for 12 consecutive weeks and were euthanized 6 mo after the first urethane injection. Mice were euthanized by cervical dislocation.

### Tissue processing: mouse lung histology and IF staining

At the time of euthanasia, lungs from the mice in each group were processed for analysis. We were unable to perform a blinded analysis in the quantification and IF analysis of the mouse tumors because the tumors in the control and experimental mice are of different numbers and sizes, and the unique class of urethane-induced large adenomas in the IKKα$^{KO}$ AT-II lung epithelial cell mice were visible by eye.

Lungs from one-third of the total number of mice of each group were harvested, and tumors on lung surfaces were counted under a Nikon SM2800 stereoscope. Another third of the lungs of each group were fixed in 10% buffered formalin overnight, transferred to 70% EtOH, or stained with X-gal (Hogan et al, 1994). In brief, whole lungs were dissected in PBS and transferred to 4% PFA for 1 h at 4°C. Samples were then rinsed in wash buffer (2 mM MgCl$_2$, 0.02% Nonidet P-40 in PBS) and incubated in lacZ stain (1 mg/ml X-gal, 200 mM K$_3$Fe$_6$, 200 mM K$_4$Fe) at 37°C overnight (Ji et al, 2006). After staining, samples were rinsed in wash buffer and post-fixed in 4% PFA overnight at 4°C. All the fixed samples were then embedded in paraffin. Five-micrometer sections were cut and stained with hematoxylin and eosin for histopathology using standard procedures and examined under a Leica DM LS2 microscope, or underwent deparaffinization and antigen retrieval using sodium citrate buffer (10 mM sodium citrate, 0.05% Tween 20, pH 6.0) before proceeding with tissue blocking for IF. In brief, primary rabbit monoclonal

HIF-1α antibody (100, NB100-479, Novus Biological or (3716 Cell Signaling) was added at a 1:100 dilution, and slides were incubated overnight at 4°C in humidified chambers. Primary antibody was washed out three times with 1× PBT, and samples were further incubated with secondary anti-rabbit GFP-conjugated antibody (Jackson Immunoresearch) at 1:500 dilution for 2 h at room temperature. Images were captured as single optical sections using an Olympus FV3000 confocal microscope (20× objective lens). The remaining one-third of the lung tumors from the rest of the mice in each group were microdissected under a Nikon SM2800 stereoscope, snap frozen in liquid nitrogen, and processed for molecular studies (RNA-seq).

### Human NSCLC tumor xenografts in immune-compromised mice

Three independent human NSCLC lines (H1437, A549, and H1299), which differ either in their p53 or K-Ras functional status were obtained from the Sanger (UK) cell line database, which provides the status of their K-Ras and p53, EGFR, ARF, and p16 alleles at the following Web sites:

H1437: https://cancer.sanger.ac.uk/cell_lines/sample/overview?id=687794.
A549: https://cancer.sanger.ac.uk/cell_lines/sample/overview?id=905949.
H1299: https://cancer.sanger.ac.uk/cell_lines/sample/overview?id=724831.

These three human NSCLC lines (IKKα$^{WT}$ and IKKα$^{KD}$) and H1299 and A549 (p52$^{WT}$ and p52$^{KD}$) were grown as tumor xenografts by subcutaneous transplantation into either side (left side for WT and right side for KD cells) of immune-compromised 5-wk-old NSG (NOD-SCID-IL2Rgamma) mice (2 × 10$^6$ cells per injection in 200 µl of PBS); and doxycycline (Sigma-Aldrich) was added to the drinking water of the mice (2 mg/ml). Mice were euthanized 3 wk (IKKα$^{WT}$ and IKKα$^{KD}$) and 4 wk (p52$^{WT}$ and p52$^{KD}$) later for tumor weight measurement and for histopathologic and molecular analysis (RNA-seq). For the RNA-seq analysis, part of the dissected tumors was immediately snap-frozen in liquid nitrogen before RNA extraction with the TRI-Reagent (Merck) protocol, according to the manufacturer's instructions.

### Cells, tissue culture, and tumor sphere growth

Human LC cells A549, H1299, and H1437 were cultured in DMEM (Sigma-Aldrich) and HEK293T and Phoenix Ampho cells in high-glucose DMEM (Sigma-Aldrich) supplemented with 10%vol/vol fetal bovine serum (Gibco; Thermo Fisher Scientific), 2 mM L-glutamine, 100 units/ml penicillin, and 100 µg/ml streptomycin (all from Seromed/Biochrom KG, Germany) at 37°C, 5% CO$_2$.

H1437 control and IKKα$^{KD}$ cells were also grown as tumor spheres in vitro under normoxia and hypoxic conditions. The IKKα$^{KD}$ experimental H1437 cells expressing the doxycycline (Dox)-inducible lentiviral IKKα shRNA-GFP cassette and the H1437 control cells stably transduced with a control lentiviral vector lacking the IKKα shRNA sequence were both verified to be GFP-positive in response to Dox using a

fluorescence microscope. Briefly, the cells were plated in ultra-low attachment 96-well plates (3474; Corning Incorporated) at a density of 1,000 cells/well in serum-free DMEM/F12 medium (D8437; Sigma-Aldrich) supplemented with 20 ng/ml epidermal growth factor (E5036; Sigma-Aldrich), 10 ng/ml basic fibroblast growth factor (F0291; Sigma-Aldrich), 5 $\mu$g/ml insulin (A11429IJ; Invitrogen), 1XB27 supplement (17504-044; Invitrogen), and 0.4% bovine serum albumin (A9576; Sigma-Aldrich) (Gu et al, 2011), and doxycycline (2 $\mu$g/ml) was included in the tumor sphere media of both the experimental and control cells. The cells were cultured for under 5% $CO_2$ at 37°C (normoxia). To induce hypoxia, the tumor spheres were placed in a hypoxic chamber with an oxygen concentration maintained at 3% $O_2$ (mild hypoxia conditions) for 7 d.

## Lentiviral/retroviral transductions

IKK$\alpha$ shRNAs were stably expressed in human NSCLC lines by lentiviral transduction. For this, a novel lentiviral vector (pFCtoG-MUbgiTnhIPW) was constructed starting from the backbone of the pFUGW lentiviral vector (kindly provided by Dr David Baltimore) (Lois et al, 2002). An miR30 expression cassette linked to GFP was placed under the control of a TetR responsive CMV promoter/enhancer. A validated human IKK$\alpha$-specific sh-oligo (Olivotto et al, 2008), which we have previously reported to produce penetrant IKK$\alpha$ knockdowns in primary human cells, was inserted in the miR30 cassette. The lentiviral vector also contains a subgenomic TetR-Ires-Puro expression cassette driven by the human ubiquitin C (UbC) gene promoter. To target TetR (Tetracycline Repressor) expression to cell nuclei, and to also provide a means for its convenient detection, a PCR-based strategy was used to fuse a HA (hemagglutinin) epitope–tagged NLS of the SV40 large T antigen to the TetR carboxy terminus (TetRNh) and a rabbit beta-globin intervening sequence derived from the pcDNA6/TR vector was inserted between the UBC promoter and the recombinant TetRNh ORF to further enhance its expression level. HEK293T cells were cultured at 70% confluency in 10 ml of complete high glucose DMEM. A mixture containing 4.5 $\mu$g lentivector pFCtoGMUbgiTnhIPW-shIKK$\alpha$, 2.0 $\mu$g of pCMV-VSVg DNA, and 5 $\mu$g of pCMV-dR8.2 DNA in 125 $\mu$l serum-free DMEM and 15 $\mu$l polyethylenimine (PEI) was first incubated for 20 min at room temperature, and then added dropwise to the growth medium of the cells, which were incubated overnight at 37°C, 5% $CO_2$. The following day, the medium was removed, the cells were washed twice with serum-free DMEM, and the transfected cells were incubated for 24 h at 37°C, 5% $CO_2$. The supernatant containing the recombinant lentiviruses was filtered through a 0.45-$\mu$M Whatman syringe filter (Sigma-Aldrich), and polybrene (Sigma-Aldrich) was added to a final concentration of 8 $\mu$g/ml. The lentivirus-containing supernatant was used to infect human LC cell lines (Stewart et al, 2003). The transduced cells were then selected in 10 $\mu$g puromycin containing complete DMEM (SantaCruz Biotechnology). IKK$\alpha$ knockdown (IKK$\alpha^{KD}$) in puromycin-resistant lentiviral expressing cells was induced by exposing the cells to doxycycline (Dox). Importantly, as a lentiviral vector control, each of the IKK$\alpha^{WT}$ human NSCLC lines were stably transduced with an empty version of the same lentiviral vector lacking the IKK$\alpha$ shRNA sequence, which (akin to the IKK$\alpha^{KD}$ cells) induces GFP expression in response to Dox. Importantly, the IKK$\alpha^{WT}$ vector control NSCLC cells were also expanded in vitro in media

supplemented with Dox, and Dox was also added to the drinking water of the control mice with the IKK$\alpha^{WT}$ NSCLC tumor xenografts.

NF-$\kappa$B p52 knockdown in human NSCLC lines was performed by transduction with an amphotyped pSuper.retro-Puro retrovirus derivative expressing a human p52 shRNA (Bernal-Mizrachi et al, 2006). To produce the infectious retrovirus, Phoenix Ampho (PhoenixA) cells at 70% confluency were transfected with 6 $\mu$g of the human shp52-puro retrovector plasmid DNA mixed in 30 $\mu$g PEI in a total volume of 150 $\mu$l of serum-free DMEM, which was incubated at room temperature for 20 min. The plasmid DNA–PEI mixture was added to the PhoenixA cells, which were incubated overnight. The next day, the medium was changed and the cells were washed twice with serum-free medium. After incubating the transfected cells in complete media for another 24 h, cellular supernatant containing the recombinant retroviruses was filtered through a 0.45-$\mu$M Whatman syringe filter (Sigma-Aldrich), and polybrene (Sigma-Aldrich) was added to a final concentration of 8 $\mu$g/ml. The retrovirus-containing supernatant was then used to infect human LC cells (Batsi et al, 2009; Sfikas et al, 2012; Markopoulos et al, 2017), and stably infected cells were selected in 10 $\mu$g puromycin.

## Western blotting antibodies

Cells from human NSCLC lines stably transduced with control and IKK$\alpha^{KD}$ lentiviral vectors or p52 shRNA–expressing retroviruses were lysed in RIPA buffer. Protein lysates of the mouse tumor samples or xenografts were extracted by Trizol (TRI-reagent; Merck) or from formalin-fixed paraffin-embedded (FFPE) tissues using the Qproteome formalin-fixed paraffin-embedded tissue kit (37623; QIAGEN) according to the manufacturer' instructions. Equal volumes of lysates from each condition were resolved by 8–10% SDS–PAGE. Immunoblotting was performed with the following primary antibodies: anti-IKK$\alpha$ (sc-87606), anti-HIF-1$\alpha$ (3716; Cell Signaling), anti-p52 (sc-298), and anti-$\beta$-actin (A5441; Sigma-Aldrich) followed by HRP-conjugated secondary antibodies. Antibody binding was detected using an ECL detection kit (Pierce ECL2 Western Blotting Substrate, 80196; Thermo Fisher Scientific).

## IF staining of cells/tumor spheres

For IF assays of HIF-1$\alpha$ protein, H1437 tumor spheres (IKK$\alpha^{WT}$ and IKK$\alpha^{KD}$) were dried briefly onto microscope slides, fixed in cold methanol:acetone (1:1) for 10 min at –20°C, and then washed three times with PBS and blocked in 1% bovine serum albumin, containing 0.1% Triton X-100 in 1× PBS (1× PBT), for 30 min. Primary rabbit polyclonal HIF-1$\alpha$ antibodies were added at the appropriate dilutions, and the slides were incubated overnight at 4°C in humidified chambers. Primary antibodies were washed out three times with 1× PBT, and the samples were further incubated with secondary anti-rabbit Cy3-conjugated antibodies for 2 h at room temperature. Tumor spheres were counterstained with 4,6-diamidino-2-phenylindole (Sigma-Aldrich) for 3 min and mounted with VECTASHIELD (Vector Laboratories). H1437 control or IKK$\alpha^{KD}$ cells were grown on coverslips and cultured for 16 h under 5% $CO_2$ at 37°C (normoxia) in complete medium supplemented with doxycycline (2 $\mu$g/ml). The cells were then exposed to 3% hypoxic conditions for 5 h. Coverslips were rinsed with PBS, fixed with cold

methanol:acetone (1:1) for 10 min at –20°C, and processed as described above for the tumor spheres. Images were captured as single optical sections using the Leica TCS-SP8 inverted confocal microscope (40× objective lens).

### Image processing and quantification of HIF-1α protein–expressing cells

All images of HIF-1α protein expressing cells were captured on an Olympus FV3000 confocal microscope or a Leica TCS SP8 inverted confocal laser scanning microscope as mentioned above. Image processing and mean intensity values of HIF-1α staining were performed using ImageJ software.

### Reverse transcription and quantitative PCR

Total RNA from mouse tumors was isolated using the TRI-Reagent (Merck) protocol, according to the manufacturer's instructions. Complementary DNA samples were prepared using PrimeScript RT Reagent Kit (Takara) and quantitative qRT-PCR reactions were performed using KAPA SYBR FAST qPCR Master Mix (Kapa Biosystems), in a Roche LightCycler 96. PCR primers for quantifying the expression levels of specific direct HIF target genes included human Slc2a1 forward TCTGGCATCAACGTGTCTTC and reverse CGATACCGGAGCCAATGGT primers; mouse Slc2a1 forward CAGTTCGGCTATAACACTGGTG and reverse GCCCCCGACAGAGAAGATG primers; and mouse HK2 forward TGATCGCCTGCTTATTCACGG and reverse AACCGCCTAGAAATCTCCAGA primers. Human and murine 18S rRNAs were reference controls. H18S: forward CTACCACATCCAAGGAAGCA and reverse TTTTTCGTCACTACCTCCCCG primers; and mouse 18S forward GTAACCCGTTGAACCCCATT and reverse CCATCCAATCGGTAGTAGCG primers. Levels of CHUK/IKKα mRNA expression in murine IKKα$^{WT}$ and IKKα$^{KO}$ lung tumors were quantified by qRT-PCR with PCR primers positioned 5′ and 3′ of murine CHUK/IKKα Exons 6 and 7, which are specifically deleted in AT-II lung epithelial cells by tamoxifen-induced CreER$^{T2}$-mediated LoxP recombination in IKKα$^{f/f}$:Sftpc-CreER$^{T2}$ mice: forward 5′ CHUK exon 6 primer GCCTCTTCTGGCAATGGAG and Reverse 3′ CHUK exon 7 primer ATAGGTCCCCCTCTCTGCTG.

### RNA sequencing and bioinformatics analysis

RNA-seq transcriptome analyses were carried out in the Greek Genome Center of the Biomedical Research Foundation Academy of Athens. Total cell RNA sample quality and integrity were verified with an Agilent RNA 6000 nano kit or also by quantifying the relative levels of expression of 28S versus 18S rRNAs by MOPS denaturing agarose gel electrophoresis. RNA-seq libraries were prepared with the Illumina TruSeq RNA v2 kit using 1 μg of total RNA. Libraries were checked with the Agilent bioanalyzer DNA1000 chip, quantified with the Qubit HS spectrophotometric method, and pooled in equimolar amounts for sequencing. 75-bp single-end reads were generated with the Illumina NextSeq500 sequencer. Raw Fastq-sequencing files were then imported into CLC Biomedical Genomics Workbench version 5.0.1. The qualities of the samples were assessed, and sequences with low quality were trimmed as follows: quality limit was set to 0.05 and ambiguous limit was set to two. Sequences with high quality were then aligned using CLC Biomedical Genomics

Workbench using the following parameters: reference type = genome annotated with genes and transcripts; reference sequence = homo_sapiens_sequence_hg38; gene track = homo sapiens Ensembl v90 genes; mRNA track = homo sapiens Ensembl v90 mRNA; mismatch cost = 2; insertion cost = 3; deletion cost = 3, length fraction = 0.8; similarity fraction = 0.8; global alignment = no; strand-specific = both; maximum number of hits for a read = 10; count paired reads as two = no; and expression value = total counts. The aligned transcriptomic expression datasets were then uploaded onto Ingenuity IPA, and Ingenuity was set to ignore features with mean expression values below 10.0 and to only consider transcripts with a $P$-value ≤ 0.05 and a fold change ≥1.5 or ≤−1.5. Ingenuity upstream analysis was then exported and are depicted as heat maps. Common differentially expressed transcripts were identified using the Ingenuity comparison tool. Briefly, ingenuity was set to compare transcript expression between human and mouse tumors. Genes that were ≥1.4 or ≤−1.4 fold increased or decreased, respectively, in two or more tumor groups were exported and depicted in heat maps using Morpheus (https://software.broadinstitute.org/morpheus).

### Statistical analyses

Statistical analyses were performed by $t$ test (two-tailed) or in some cases by a one-way ANOVA statistical test (as indicated in specific figure legends). Differences with $P < 0.05$ were considered statistically significant. Data were represented as mean ± SD.

### Data availability

The RNA sequencing data can be accessed on the NCBI GEO database with GEO Access number GSE140432.

# Supplementary Information

# Acknowledgements

KB Marcu thanks Ms Laurie Levine for expert help with mouse breeding, Ms Juei-Suei Chen for PCR-based mouse genotyping, and Prof David Baltimore for providing the pFUGW lentiviral vector. We also gratefully acknowledge Prof Brigid Hogan for her gift of the Sftpc-CreER$^{T2}$ mice. E Chavdoula and KB Marcu gratefully acknowledge the kind hospitality of Dr Philip Tsichlis for allowing E Chavdoula to complete the immunofluorescence and Western blot experiments in his laboratory at the Ohio State University Comprehensive Cancer Center. E Chavdoula acknowledges the assistance of Dr Evangelia Xingi in the Light Microscopy facility of the Hellenic Pasteur Institute. This research was in part co-financed by European Union-European Science Foundation and the Greek national funds through the Operational Program "Education and Lifelong Learning" of NSRF (National Research Foundation)–Research Funding Program THALIS2007-2013 (MIS379435) (E Kolettas, A Kokkalis, D Thanos, KB Marcu), a project grant "Advanced Research Activities in Biomedical and Agro alimentary Technologies" (MIS5002469) of the Operational Programme "Competitiveness, Entrepreneurship and Innovation" (NSRF2014-20, EU-ERDF) (E Kolettas), and a research grant in Biomedical Sciences and from Fondation Santé (E Kolettas). E Chavdoula and GS Markopoulos were supported by a Thalis grant, and E Vasilaki by a

Advanced Research Activities in Biomedical Technologies and a Fondation Santé grant. The funders had no role in the study design, data collection and analysis, decision to publish, or preparation of the manuscript. A Kokkalis was supported by the KMW offsets program.

## Author Contributions

E Chavdoula: data curation, formal analysis, validation, investigation, visualization, methodology, and writing—original draft.

DM Habiel: resources, data curation, software, formal analysis, investigation, visualization, methodology, and writing—review and editing.

E Roupakia: data curation, formal analysis, validation, investigation, visualization, and methodology.

GS Markopoulos: data curation, formal analysis, validation, investigation, visualization, and methodology.

E Vasilaki: resources, investigation, and methodology.

A Kokkalis: resources, investigation, visualization, and methodology.

AP Polyzos: resources, data curation, software, formal analysis, validation, investigation, and methodology.

H Boleti: resources, supervision, funding acquisition, project administration, and writing—review and editing.

D Thanos: resources, software, validation, investigation, and methodology.

A Klinakis: resources, supervision, investigation, visualization, methodology, and project administration.

E Kolettas: conceptualization, supervision, funding acquisition, project administration, and writing—review and editing.

KB Marcu: conceptualization, supervision, investigation, methodology, and writing—original draft and project administration.

## Conflict of Interest Statement

The authors declare that they have no conflict of interest.

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
