## [Reviewer comments · Life Science Alliance]

Life Science Alliance

CHUK/IKK α loss in lung epithelial cells enhances NSCLC growth associated with HIF up-regulation

Evangelia Chavdoula, David M. Habel, Eugenia Roupakia, Georgios S. Markopoulos, Eleni Vasilaki, Antonis Kokkalis, Alexander P. Polyzos, Haralabia Boleti, Dimitris Thanos, Apostolos Klinakis, Evangelos Kolettas, and Kenneth B. Marcu

DOI: <https://doi.org/10.26508/lsa.201900460>

Corresponding author(s): Kenneth Marcu, Stony Brook University

Review Timeline:

Submission Date:	2019-06-19
Editorial Decision:	2019-07-19
Revision Received:	2019-10-31
Editorial Decision:	2019-11-05
Revision Received:	2019-11-15
Editorial Decision:	2019-11-18
Revision Received:	2019-11-18
Accepted:	2019-11-19

Scientific Editor: Andrea Leibfried

Transaction Report:

July 19, 2019

Re: Life Science Alliance manuscript #LSA-2019-00460-T

Prof. Kenneth B Marcu
Stony Brook University
Biochemistry and Cell Biology Dept. Life Sciences Bldg. Rm. 330 Stony Brook University
Stony Brook, New York 11794-5215

Dear Dr. Marcu,

Thank you for submitting your manuscript entitled "CHUK/IKK α in lung epithelial cells functions as a tumor suppressor in two independent in vivo models of non-small-cell lung cancer" to Life Science Alliance. The manuscript was assessed by expert reviewers, whose comments are appended to this letter.

As you will see, the reviewers support publication of a revised version of your manuscript in Life Science Alliance. We would thus like to invite you to submit a revised version, addressing all reviewer comments. Importantly, a few controls are needed as well as restructuring the manuscript and providing more information. We think it is straightforward to address these issues, but please do get in touch in case you would like to discuss individual revision points further.

Thank you for this interesting contribution to Life Science Alliance. We are looking forward to receiving your revised manuscript.

Sincerely,

Andrea Leibfried, PhD
Executive Editor

Life Science Alliance
Meyerhofstr. 1
69117 Heidelberg, Germany
t +49 6221 8891 502
e a.leibfried@life-science-alliance.org
www.life-science-alliance.org

B. MANUSCRIPT ORGANIZATION AND FORMATTING:

Reviewer #1 (Comments to the Authors (Required)):

This manuscript investigates the role of IKK α in in vivo models of non-small-cell lung cancer (NSCLC). As noted by the authors other studies have previously examined IKK α in this context but with differing conclusions as to whether it plays a tumor promoting or suppressing role. A key difference with this study is that the authors have chosen a mouse model that permits deletion of IKK α in alveolar type II lung epithelial cells. In addition to this they have employed a panel of three NSCLC cell lines to compare to the data obtained from the mouse data.

In all cases they find they IKKalpha functions as a tumor suppressor in this context. This is in contrast to IKKbeta, which drives activation of the canonical NF-kB pathway and functions as a tumor promoter in this same mouse model. Moreover, this function of IKKalpha appears to be independent of its ability to drive activation of the non-canonical NF-kB pathway. An important finding in this paper is that loss of IKKalpha results in upregulation of HIF dependent hypoxia signaling, providing a potential explanation for the tumor suppressor role they observe.

Overall I found this to be a clearly written manuscript. The conclusions drawn from the data are justified and it makes an important contribution to this research area. I have some minor comments about how the manuscript could be improved.

(1) I think it would be better if all the graphs were presented as scatter plots as in Figure 3 (right panel). This gives a better understanding of the variability of the data sets. These problems with using bar charts to present data are discussed in an excellent article by Weissgerber et al in PLoS Biology (<https://doi.org/10.1371/journal.pbio.1002128>).

(2) With the quantification of tumor numbers (e.g. Figure 2) or immunofluorescence/IHC (e.g. Figure 5) the authors should indicate in the methods or figure legends if the samples were blinded before analysis.

(3) In the discussion, the authors indicate that IKKalpha was nuclear in the cell lines (Discussion, page 10, data not shown) but in Figure 5, it appears to be mainly cytoplasmic. Can the authors comment on this?

(4) Can the authors perform a western blot for HIF1a from either the tumor samples or cell lines to confirm and support their data from IF indicating loss of IKKalpha leads to HIF1a upregulation?

(5) In Figure 8, can the authors indicate how many control IKKalpha wild type samples were used for this analysis?

(6) I noticed that there were big differences between the genes listed in Figure 2 and those identified as a HIF1a gene signature in Tables 1 and S1. Indeed there were major differences between Tables 1 and S1 themselves. I think it would be useful and informative to also provide a heatmap as in Figure 2 but showing the HIF1a signature genes from Tables 1 and S1. While it might be expected that the HIF1a signature may vary depending on context I think this heatmap would help the reader better interpret this data. It would also help if the authors could discuss these differences in the text.

(7) On page 4, SRD needs to be defined.

Reviewer #2 (Comments to the Authors (Required)):

This manuscript focuses on Investigating the specific function of IKKalpha and beta subunits in tumorigenesis in the lung, which is extremely relevant. Thus, from this perspective I find the work relevant. However, there are several issues that should be revised to improve the quality of the data and the clarity of the message. Importantly, authors need to better describe the experiments performed and the results obtained, not only in the figure legends section, but also in the text. This

applies to the whole body of the manuscript. For example, in several sections the results obtained from a set of experiments are mentioned with a short sentence followed by "results are shown in figure X", without any further description of the observations.

In general, not only in the results section, but also, the text is very difficult to follow and the writing should be extensively revised for clarity. I could provide examples but this observation is extensive to almost all sections.

I also find surprising that almost every figure is composed of a single panel instead of subfigures, as it is the usual format of most journals.

Other concerns:

In the experiments that include inducible KD tumors (IKKalpha, p52, etc), the extent of the reduction have to be tested in the tumors at the end of the experiment (IHC, WB or PCR).

Levels of HIF1-alpha in response to IKKalpha variations need to be confirmed by WB.

The paper from Rius and colleagues investigating the role of IKKbeta and NFkappaB in HIF-1 alpha induction by hipoxia needs to be cited and discussed, as well as the work from the Rocha's group on NFkappaB in hypoxia.

The 2 reviewers concerns and suggestions required us to perform a number of additional control experiments in the form of multiple western/immunoblots. Our postdoctoral fellow and the first author of our paper (Dr. Ela Chavdoula) needed almost 3 months to complete these additional experiments mostly because about 8 months ago she had moved to a new senior postdoctoral position at the Comprehensive Cancer Center in Columbus, Ohio (USA) and frozen human NSCLC tumor xenografts and mouse lung paraffin sections had to be shipped to her from the BRFAA Institute in Athens, Greece where she had performed all of the *in vivo* experiments that were presented in our original submitted manuscript. Thus almost all of the figures in our original paper have been extensively revised which resulted in almost all of them now being multi-panel figures and one additional text figure was also added to our revised paper as well. I am pleased to say that all of these many, additional quantitative western blots have confirmed and greatly strengthened the major novel conclusion of our paper: “The knockout of IKKalpha/CHUK in the AT-II lung epithelial cells of urethane-induced lung cancer in mice and the knockdown of IKKalpha/CHUK in the Human H1437 NSCLC line grown as tumor xenografts in immune compromised mice both result in the up-regulation of HIF-1alpha protein and activation of its target genes which helps to explain their enhanced growth under hypoxic conditions *in vivo*.”

Our Point by point replies to each of the 2 reviewers comments.

Reviewer #1 (Comments to the Authors (Required)):

This manuscript investigates the role of IKKalpha in *in vivo* models of non-small-cell lung cancer (NSCLC). As noted by the authors other studies have previously examined IKKalpha in this context but with differing conclusions as to whether it plays a tumor promoting or suppressing role. A key difference with this study is that the authors have chosen a mouse model that permits deletion of IKKalpha in alveolar type II lung epithelial cells. In addition to this they have employed a panel of three NSCLC cell lines to compare to the data obtained from the mouse data.

In all cases they find they IKKalpha functions as a tumor suppressor in this context. This is in contrast to IKKbeta, which drives activation of the canonical NF-kB pathway and functions as a tumor promoter in this same mouse model. Moreover, this function of IKKalpha appears to be independent of its ability to drive activation of the non-canonical NF-kB pathway. An important finding in this

paper is that loss of IKKalpha results in upregulation of HIF dependent hypoxia signaling, providing a potential explanation for the tumor suppressor role they observe. Overall I found this to be a clearly written manuscript. The conclusions drawn from the data are justified and it makes an important contribution to this research area. I have some minor comments about how the manuscript could be improved.

Our reply: “We very much appreciate the reviewers kind overall remarks about the importance of our findings and novelty of our conclusions; and we have revised our manuscript to comply with each of the reviewer’s concerns and suggestions to the best of our ability.”

(1) I think it would be better if all the graphs were presented as scatter plots as in Figure 3 (right panel). This gives a better understanding of the variability of the data sets. These problems with using bar charts to present data are discussed in an excellent article by Weissgerber et al in PLoS Biology (<https://doi.org/10.1371/journal.pbio.1002128>).

Our reply: “We appreciate the reviewer’s comment about the bar graphs in our paper and agree that scatter plots would be better shown for the mouse tumors since we are dealing there with variables in each animal wherein the tumors develop and grow over a 6 month time span; and so in revised versions of Figures 1, 2 and 7 showing these specific results we have replaced the original bar graphs with new scatter plots. However, bar graphs for the levels of IKKalpha protein in the WT control mouse lung or in the lungs of the experimental mice have nothing to do with tumor development since the targeted AT-II IKKalpha deletions were induced by tamoxifen treatment just prior to exposing the mice Urethane to produce lung adenomas. In addition, since all of the remaining bar graphs are for our data with the clonal Human H1437, A549 and H1299 NSCLC cell lines, which were only grown *in vivo* as tumor xenografts, (starting with an inoculum of $\sim 2 \times 10^6$ cells), in immune compromised NSG mice for a short time span (~ 3 weeks) or in the case of H1437 IKKalpha^{WT} and IKKalpha^{KD} cells, which were also maintained *in vitro* under hypoxic conditions, in our opinion bar graphs should be a perfectly reasonable way to present the IKK^{KD} and HIF-1alpha results obtained with these cells.”

(2) With the quantification of tumor numbers (e.g. Figure 2) or immunofluorescence/IHC (e.g. Figure 5) the authors should indicate in the methods or figure legends if the samples were blinded before analysis.

Our reply: “Sorry but we were unable to perform a blinded analysis in the quantification of the tumor numbers and the IF images since we are dealing with tumors of different numbers and sizes. For example, in the case of the IF stainings, in the lung sections of the experimental AT-II IKKalpha KO mice, the large tumors were visible even without the use of a microscope. However, we assure

the reviewer that we have presented all the results in our paper in a completely unbiased manner. This has now be so stipulated in the methods and appropriate figure legends of the revised version of our paper."

(3) In the discussion, the authors indicate that IKKalpha was nuclear in the cell lines (Discussion, page 10, data not shown) but in Figure 5, it appears to be mainly cytoplasmic. Can the authors comment on this?

Our reply: "Sorry, but we are confused by the reviewer's comment #3 for two reasons: (1) The original Figure 5 of our paper has no cells in it and (2) more importantly, there are no figures in the paper showing the intracellular localization of IKKalpha. The only figures in which cells are shown are in some of the Immunofluorescence results for the quantification of HIF-1alpha which are now only in Figure 7. Moreover, the predominantly nuclear localization of IKKalpha in the nuclei of the Human NSCLC lines (data not shown in our paper) is not a very surprising observation as it agrees with many other published papers that IKKalpha has an active nuclear localization signal, which facilitates its nuclear retention in transformed cells with pro-inflammatory activated canonical NFkappaB."

(4) Can the authors perform a western blot for HIF1a from either the tumor samples of cell lines to confirm and support their data from IF indicating loss of IKKalpha leads to HIF1a up-regulation?

Our reply: "In the revised version of our paper we have now included several HIF-1alpha Western blot results, which indeed clearly confirm our IF analysis showing its up-regulation in the IKKalpha^{KO} mouse lung adenomas and in the IKKalpha^{KD} H1437 tumor xenografts and cells grown in vitro under hypoxia."

(5) In Figure 8, can the authors indicate how many control IKKalpha wild type samples were used for this analysis?

Our reply: "A total of 3 independent IKKalpha^{WT} control mouse lungs containing small adenomas were used for the QRTPCR analysis of HIF-1alpha target genes in 5 different IKKalpha^{KO} large adenomas. Three control mice were used because on the basis of their comparable WT endogenous IKKalpha mRNA expression levels they clustered well in the Ingenuity analysis that identified the HIF-1alpha pathway as being up-regulated in the large IKKalpha^{KO} lung adenomas. "

(6) I noticed that there were big differences between the genes listed in Figure 2 and those identified as a HIF1a gene signature in Tables 1 and S1. Indeed there were major differences between Tables 1 and S1 themselves. I think it would be useful and informative to also provide a heatmap as in Figure 2 but showing the HIF1a signature genes from Tables 1 and S1. While it might be expected that the

HIF1a signature may vary depending on context I think this heatmap would help the reader better interpret this data. It would also help if the authors could discuss these differences in the text.

Our reply: "The reviewer must be referring to the original Heat Map in Figure S2 (not Figure 2). We appreciate the reviewers comment and in a new Figure 6 of our revised paper we have included 2 new heat maps comparing the HIF1alpha signature genes in the mouse large adenomas and in the IKKalpha KD H1437 tumor xenografts. Since these 2 heat maps contain all of the information that was presented in Tables I and S1 of our original manuscript we have also removed these two Tables from the revised version of our paper."

(7) On page 4, SRD needs to be defined.

Our reply: "Sorry about that. SRD is an abbreviation for Signal Response Domain which has now been indicated in the Introduction of our revised manuscript."

Reviewer #2 (Comments to the Authors (Required)):

This manuscript focuses on Investigating the specific function of IKKalpha and beta subunits in tumorigenesis in the lung, which is extremely relevant. Thus, from this perspective I find the work relevant. However, there are several issues that should be revised to improve the quality of the data and the clarity of the message. Importantly, authors need to better describe the experiments performed and the results obtained, not only in the figure legends section, but also in the text. This applies to the whole body of the manuscript. For example, in several sections the results obtained from a set of experiments are mentioned with a short sentence followed by "results are shown in figure X", without any further description of the observations. In general, not only in the results section, but also, the text is very difficult to follow and the writing should be extensively revised for clarity. I could provide examples but this observation is extensive to almost all sections.

Our reply: "Sorry to hear that the reviewer thought that we did not describe the results well in the manuscript text and that it was also difficult to read. To comply with the reviewer's concerns in this regard we have extensively revised the results section of the manuscript to as best as we can describe in clear detail all of the findings in each Figure and now also hope that the text is a bit easier to follow in spite of how much data is in the paper about both *in vivo* NSCLC models."

I also find surprising that almost every figure is composed of a single panel instead of subfigures, as it is the usual format of most journals.

Our reply: "In our revised manuscript we have added a number of additional panels to almost every text figure, which was also now necessary in the revised version of

our paper since we have now added a lot of additional Western blot data for HIF1alpha, IKKalpha and p52 throughout the manuscript. "

Other concerns:

In the experiments that include inducible KD tumors (IKKalpha, p52, etc), the extent of the reduction have to be tested in the tumors at the end of the experiment (IHC, WB or PCR). Levels of HIF1-alpha in response to IKKalpha variations need to be confirmed by WB.

Our reply: "Okay, to comply with the reviewer's concern and suggestion we have added a number of Western blots to the revised paper as follows: (1) Quantified Western blots showing the loss of IKKalpha protein in the lungs of the IKKalpha(f/f) mice in response to tamoxifen, (2) Knock-down of NFkappaB p52 in the processed tumor xenografts, (3) Knock-down of IKKalpha in the H1437-IKKalpha^{KD}, A549-IKKalpha^{KD} and H1299-IKKalpha^{KD} tumor xenografts, (3) Higher levels of HIF1alpha in multiple AT-II IKKalpha^{KO} mouse large mouse adenomas are now also confirmed by western blotting and (4) Quantified western blots also now showing higher levels of HIF-1alpha in multiple independent H1437 IKKalpha^{KD} tumor xenografts and in H1437 IKKalpha^{KD} cells exposed to hypoxia."

The paper from Rius and colleagues investigating the role of IKKbeta and NFkappaB in HIF-1 alpha induction by hypoxia needs to be cited and discussed, as well as the work from the Rocha's group on NFkappaB in hypoxia.

Our reply: "Many thanks for pointing this out which was an oversight on our part. We have added these specific references towards the end of the discussion section of our revised paper. We have discussed these prior findings on how the IKKbeta-dependent canonical NFkappaB pathway directly activates HIF-1alpha gene expression in lung cancer in the context of our new findings showing that in contrast to the well known effects of IKKbeta, IKKalpha loss (independent of NFkappaB) can up-regulate the levels of activated HIF1alpha protein in both mouse and human *in vivo* NSCLC models."

In conclusion we believe that we have replied as best as we can to each of the 2 reviewers concerns and that our extensively revised manuscript now meets with the approval of the reviewers and the LSA editors. Moreover, with the additional new quantitative western blot results that we have performed, we believe that the major take home message of our manuscript has been significantly strengthened; and thus we have also modified the title of our manuscript to better reflect that novel content: "CHUK/IKKalpha loss in lung epithelial cells contributes to the development and growth of non-small-cell-lung cancer (NSCLC) in conjunction with hypoxia inducible pathway up-regulation."

November 5, 2019

Re: Life Science Alliance manuscript #LSA-2019-00460-TR

Prof. Kenneth B Marcu
Stony Brook University
Biochemistry and Cell Biology Dept. Life Sciences Bldg. Rm. 330 Stony Brook University
Stony Brook, New York 11794-5215

Dear Dr. Marcu,

Thank you for submitting your revised manuscript entitled "CHUK/IKKalpha loss in lung epithelial cells enhances NSCLC growth associated with HIF up-regulation" to Life Science Alliance. I now assessed your revised manuscript and your point-by-point response to the concerns previously raised.

While I think that many of the reviewer concerns are adequately addressed, a few remain in my view and still need your attention. I would thus like to invite you to further revise your manuscript, addressing the following:

- The analyses of the RNA-seq results is difficult to follow.
 - o You first mention heat maps for IKKalphaKO and IKKalphaKD adenomas/tumors showing few genes with similar up- or down-modulations; => these maps should be shown (or excel spreadsheets provided).
 - o You then discuss the comparative analysis shown in Fig S2. The color key ranging from 0 to 1 for this analysis does not make sense in my view - please change.
 - o You move on to the Ingenuity upstream regulator analysis. It is unclear to me how you determined direct HIF1alpha target genes for this analysis - please add this information. I also think that subsequently highlighting HIGD2A does not make sense in absence of knowledge on HIGD2A being a direct HIF1a target or not and in absence of this DEG in the lists of Fig 6
 - o I think the reviewer's concern regarding the DEGs listed ("...big differences between the genes listed in Figure S2 and those identified as HIF1a gene signatures.... It would also help if the authors could discuss these differences in the text.") needs to get better addressed: please discuss in more detail that the results seem to be rather context dependent, leading to little overlap
 - o Along these lines, HK2 and Slc2a1 are not significantly DEGs in the lists presented across all analyses (HK2 listed in 6A but not 6B; SLC2A1 in 6B but not A; only SLC2A1 in S2); this should get discussed

- Also, please:

- o deposit the RNA-seq data in a repository
- o streamline the figure legends - these are very hard to read. I would suggest to call out each panel with a capitalized letter first, followed by the description; please also make sure to call out all panels within the manuscript text itself
- o please further streamline figures by adding more panel descriptors; eg., change figure 1 to having panels A-J; introduce panel labels for figure 3; introduce separate panel labels for tumors and western blots in figure 5; introduce separate panel labels for western blot and IF images in figure 7; introduce separate panel labels for western blots in figure 9; introduce separate panel labels for IF

- images and quantification in figure 10; introduce separate panel labels for figure S3)
- o please note that some scale bars, arrows, and fonts are blurry at the moment, please increase the resolution of your figures
- o please add the statistical test used to each figure legend next to the p-values mentioned (missing in Fig2, Fig8, FigS1, FigS4)
- o please mention error bars in the legend of FigS3

Thank you for this interesting contribution to Life Science Alliance. We are looking forward to receiving your revised manuscript.

Sincerely,

B. MANUSCRIPT ORGANIZATION AND FORMATTING:

We encourage our authors to provide original source data, particularly uncropped/-processed

electrophoretic blots and spreadsheets for the main figures of the manuscript. If you would like to add source data, we would welcome one PDF/Excel-file per figure for this information. These files will be linked online as supplementary "Source Data" files.

Dear Andrea,

I have uploaded our revised manuscript again today, which has been extensively modified in the ways that you recommended! My replies back to the 1st reviewer have also been modified to include more detailed explanations about the RNA seq data and how Ingenuity bioinformatic analysis curates gene signatures to predict affected cellular pathways. I would like to personally thank you very much for taking the time to go through my initial revision in such detail, for your very helpful suggestions on how in your opinion we can improve our revised submission.

Our targeted point by point replies to each of your helpful suggestions, which also explain how we have modified our revised paper, are as follows:

o You first mention heat maps for IKK α KO and IKK α KD adenomas/tumors showing few genes with similar up- or down-modulations; => these maps should be shown (or excel spreadsheets provided).

o You then discuss the comparative analysis shown in Fig S2. The color key ranging from 0 to 1 for this analysis does not make sense in my view - please change.

Our reply: The heat maps that were referenced are present in Figure S2. Indeed, the key and numbers within the heat maps are not well described in our figure legend. This heat map is showing relative expression of the indicated transcript in the various tumors, where 1 or -1 are indicating high or low relative expression of this transcript in one tumor relative to the other tumors. The numbers in each heat-map square indicates the fold change of expression in the IKK α ^{KO} relative to WT tumors. Given that this is confusing and the main purpose of this figure is to summarize common DEGs, we have modified this heat map to show absolute expression (were the numbers will match the key) in the revised manuscript.

o You move on to the Ingenuity upstream regulator analysis. It is unclear to me how you determined direct HIF1 α target genes for this analysis - please add this information. I also think that subsequently highlighting HIGD2A does not make sense in absence of knowledge on HIGD2A being a direct HIF1 α target or not and in absence of this DEG in the lists of Fig 6.

Our reply: Ingenuity upstream analysis is based on manually curated content part of Ingenuity's knowledge base (i.e. each one of these downstream targets are based on published or bioinformatically mined connections). Ingenuity upstream analysis then looks at downstream transcripts that were up-regulated in the analyzed datasets and infers activation of upstream regulators that have been shown to modulate these transcripts. It is with this tool we were able to identify the potential involvement of the HIF pathway, which we subsequently validated experimentally. Further, HIGD2A has indeed been reported to be a direct HIF-1 α target gene (Ameri, K., et al. 2013. Nuclear Localization of the Mitochondrial Factor HIGD1A during Metabolic Stress. *PLOS ONE* 8:e62758); and thus we prefer to indicate that it is the one up-regulated DEG that is shared in common in our two *in vivo* IKK α loss NSCLC models. We have also included a much more detailed explanation of how Ingenuity analysis works in the manuscript text as follows:

"To determine if specific pathways associated with tumor development and growth could be altered in response to IKK α loss, we employed Ingenuity upstream regulator analysis, which infers upstream regulator activation based on differentially expressed transcripts, using ingenuity's manually curated knowledge base (QIAGEN Inc., <https://www.qiagenbio-informatics.com/products/ingenuity-pathway-analysis>). This analysis predicted the activation of HIF-1 α in the mouse urethane-induced large IKK α ^{KO} lung adenomas (Table 1) and in the IKK α ^{KD} H1437 tumor xenografts (Table S2). Interestingly, the one up-regulated DEG that is shared in common between the murine IKK α ^{KO} large lung adenomas and the H1437, A549 and H1299 IKK α ^{KD} human NSCLCs is HIGD2A (hypoxia inducible domain family member 2A), encoding a cytochrome c oxidase complex (complex IV) subunit. HIGD2A is the terminal enzyme in the mitochondrial respiratory chain that is regulated by HIF1A (Ameri, K., et al. 2013. Nuclear Localization of the Mitochondrial Factor HIGD1A during Metabolic Stress. *PLOS ONE* 8:e62758) and has been shown (akin to the related protein HIGD1A) to enhance cell survival under hypoxia (An et al., 2011; Salazar et al., 2019)."

o I think the reviewer's concern regarding the DEGs listed ("...big differences between the genes listed in Figure S2 and those identified as HIF1 α gene signatures.... It would also help if the authors could discuss these differences in the text.") needs to get better addressed: please discuss in more detail that the results seem to be rather context dependent, leading to little overlap

o Along these lines, HK2 and Slc2a1 are not significantly DEGs in the lists presented across all analyses (HK2 listed in 6A but not 6B; SLC2A1 in 6B but not A; only SLC2A1 in S2); this should get discussed.

Our reply: Thanks very much for these excellent suggestions. We have modified part of the discussion section of our manuscript to now include a much more detailed explanation about why there are “big differences” between the Heat Maps in our revised text Figure 6 and in the revised version of the DEGs in Figure S2.

“Ingenuity upstream analysis predicted the activation of the HIF-1 α upstream regulator based on the enrichment of differentially expressed transcripts in our transcriptomic analysis, which overlap with ingenuity’s knowledge base for HIF-1 α downstream targets. As this HIF-1 α target gene signature is manually curated from a variety of independent studies using different cell types and tissues, it is composed of a mixture of HIF-1 α signatures. Moreover, because several studies have indicated that HIF-1 α induced genes may differ in different cell types and cancer cells with diverse mutations (Benita et al., 2009; Dengler et al., 2014), HIF-1 α activation may have led to the activation of different downstream target genes in our two independent NSCLC *in vivo* models. In our experiments enhanced HIF-1 α protein expression and activity in the absence of IKK α was most apparent in the murine urethane-induced NSCLC *in vivo* model and in human H1437 IKK α ^{KD} NSCLC cells grown as tumor xenografts or maintained *in vitro* either as tumor spheres or monolayer cells under hypoxic conditions. Moreover, activation of HIF-1 α in the H1437 IKK α ^{KD} cells was also correlated with their more robust tumor burden as tumor xenografts compared to IKK α ^{KD} H1299 and A549 xenografts. However, there were few HIF-1 α target genes shared amongst the up-regulated DEGs in the absence of IKK α in our mouse urethane NSCLC model versus three unrelated human NSCLCs (H1437, A549 and H1299) grown *in vivo* as tumor xenografts, in agreement with previously published findings (Benita et al., 2009; Dengler et al., 2014) and thus suggesting that there is potential variability in the HIF-1 α target gene signatures of unrelated NSCLC tumors with different activating mutations. To validate the role of HIF-1 α in our two NSCLC models, we performed experiments to assess HIF-1 α activity using known downstream HIF-1 α direct target genes. Indeed, SLC2A1 (which was up-regulated in our RNAseq analysis as shown in Figure S2A&B) and HK2 (which was not amongst the DEGs in our RNAseq analysis) were both found to be significantly up-regulated in qRT-PCR analysis of the mouse IKK α ^{KO} urethane-induced large lung adenomas (Figure 8A&B); and SLC2A1 was also found to be enhanced in the IKK α ^{KD} H1437 tumor xenografts (Figure S4A). This discrepancy in HK2 expression may be due to differences in the techniques used to assess the expression of this transcript (e.g., RNAseq vs. qRT-PCR) or the depth of coverage in our RNAseq results. Overall, we believe that taken together our quantitative HIF-1 α protein and qRT-PCR analysis of selected direct HIF-1 α target genes suggest that IKK α loss can promote the activation of HIF-1 α in NSCLC growth.”

o deposit the RNA-seq data in a repository

Our Reply: We have deposited the FASTQ RNA Seq. raw data files, the processed bioinformatic data files and Metadata worksheets on the NCBI GEO database with GEO Access# GSE140432 and this has now been indicated at the end of the paper’s Methods section.

o streamline the figure legends - these are very hard to read. I would suggest to call out each panel with a capitalized letter first, followed by the description; please also make sure to call out all panels within the manuscript text itself.

Our Reply: As suggested I have greatly streamlined the Figure Legend calling out each subpanel in Boldface type at the beginning of each legend followed by descriptions of each in order and have called out all of the panels in the MS text as well.

o please further streamline figures by adding more panel descriptors; eg., change figure 1 to having panels A-J; introduce panel labels for figure 3; introduce separate panel labels for tumors and western blots in figure 5; introduce separate panel labels for western blot and IF images in figure 7; introduce separate panel labels for western blots in figure 9; introduce separate panel labels for IF images and quantification in figure 10; introduce separate panel labels for figure S3).

Our Reply: I have modified Figure 1 exactly as you recommended and have introduced more panels in Figure 3 and separate panels for results in Figures 5, 7, 9, 10 and Figure S3.”

o please note that some scale bars, arrows, and fonts are blurry at the moment, please increase the resolution of your figures

Our Reply: The TIFFs that I had originally uploaded were compressed TIFFs (~200 kb in size), which is what likely caused the problems that you mentioned. This time I will be uploading with the MS text high resolution TIFFs (each being ~2 MB in size).

o please add the statistical test used to each figure legend next to the p-values mentioned (missing in Fig2, Fig8, FigS1, FigS4). Please mention error bars in the legend of FigS3

Our Reply: The specific Stat tests used are now next to the P values in the relevant Figure legends; and we have made mention of the error bars in FigS3's legend as well.

Many thanks again Andrea for your very detailed suggestions on how to better improve our manuscript as per the original reviewers comments and concerns. We hope that our extensively revised manuscript now meets with their your approval.

Best regards, Ken"

Kenneth B. Marcu
Emeritus Professor

November 18, 2019

RE: Life Science Alliance Manuscript #LSA-2019-00460-TRR

Prof. Kenneth B Marcu
Stony Brook University
Biochemistry and Cell Biology Dept. Life Sciences Bldg. Rm. 330 Stony Brook University
Stony Brook, New York 11794-5215

Dear Dr. Marcu,

Thank you for submitting your revised manuscript entitled "CHUK/IKKalpha loss in lung epithelial cells enhances NSCLC growth associated with HIF up-regulation". I appreciate your response and the introduced changes and would be happy to publish your paper in Life Science Alliance. Before sending you an official acceptance letter, please check your figures one more time, please. Some of them still have text/arrows etc that are not of production quality - please improve.

A. FINAL FILES:

B. MANUSCRIPT ORGANIZATION AND FORMATTING:

Thank you for your attention to these final processing requirements.

Sincerely,

Andrea Leibfried, PhD
Executive Editor
Life Science Alliance
Meyerohofstr. 1
69117 Heidelberg, Germany
t +49 6221 8891 502
e a.leibfried@life-science-alliance.org
www.life-science-alliance.org

November 19, 2019

RE: Life Science Alliance Manuscript #LSA-2019-00460-TRRR

Prof. Kenneth B Marcu
Stony Brook University
Biochemistry and Cell Biology Dept. Life Sciences Bldg. Rm. 330 Stony Brook University
Stony Brook, New York 11794-5215

Dear Dr. Marcu,

Thank you for submitting your Research Article entitled "CHUK/IKK α loss in lung epithelial cells enhances NSCLC growth associated with HIF up-regulation". It is a pleasure to let you know that your manuscript is now accepted for publication in Life Science Alliance. Congratulations on this interesting work.

DISTRIBUTION OF MATERIALS:

Again, congratulations on a very nice paper. I hope you found the review process to be constructive and are pleased with how the manuscript was handled editorially. We look forward to future exciting submissions from your lab.

Sincerely,
